# Agricultural Applications of Superabsorbent Polymer Hydrogels

**DOI:** 10.3390/ijms232315134

**Published:** 2022-12-01

**Authors:** Elena L. Krasnopeeva, Gaiane G. Panova, Alexander V. Yakimansky

**Affiliations:** 1Institute of Macromolecular Compounds, Russian Academy of Sciences, St. Petersburg 199004, Russia; 2Agrophysical Research Institute, Russian Academy of Sciences, St. Petersburg 195220, Russia

**Keywords:** superabsorbent polymers, hydrogels, polyacrylates, polysaccharides, agriculture, water retention, nutrient release, soil amendment, plant germination, plant growth

## Abstract

This review presents data from the past five years on the use of polymeric superabsorbent hydrogels in agriculture as water and nutrient storage and retention materials, as well as additives that improve soil properties. The use of synthetic and natural polymeric hydrogels for these purposes is considered. Although natural polymers, such as various polysaccharides, have undoubted advantages related to their biocompatibility, biodegradability, and low cost, they are inferior to synthetic polymers in terms of water absorption and water retention properties. In this regard, the most promising are semi-synthetic polymeric superabsorbents based on natural polymers modified with additives or grafted chains of synthetic polymers, which can combine the advantages of natural and synthetic polymeric hydrogels without their disadvantages. Such semi-synthetic polymers are of great interest for agricultural applications, especially in dry regions, also because they can be used to create systems for the slow release of nutrients into the soil, which are necessary to increase crop yields using environmentally friendly technologies.

## 1. Introduction

Agriculture and food production are key sectors for sustainable development. To control the production process of plants efficiently, realizing the genetically determined potential of productivity, it is necessary to optimize the conditions of the environment and anthropogenic factors and to maintain them in the optimal ranges [1]. In order to achieve these aims, one needs not only to maximize the crop yield per unit area through the ever-increasing use of fertilizers and pesticides but also to cope with such harmful effects of their excessive use as soil degradation, water pollution, climate changes, and plant deceases [2].

One of the most promising ways of solving these problems is related to the use of modern nanotechnologies not only to optimize light, root-inhabited, and air environments for the growth, development of plants, and the realization of their production but also to detect and eliminate or neutralize pathogens and potentially dangerous agricultural chemicals [3]. In this respect, soil analogs are rather important, especially in regions with poor soils, especially in the Arctic. Thus, we studied the effect of thin-film soil analogs based on the suspensions of such amendments as Cambrian clay and /or sapropel in different ratios on the production process of lettuces [4], cucumbers [5], and tomatoes [6]. In comparison with a low-volume soil analog based on high-moor peat of a low degree of decomposition “Agrophyte” (control group), an increase in such growth indicators of lettuces as the height and number of leaves, wet and dry mass of plants, net productivity of photosynthesis was found. Possible reasons for the lower productivity of lettuce plants in the control group were found to be due to an increase in water intake in leaf tissues without additional mineral and/or organic nutrition [4]. Accelerated growth of the cucumber hybrid Tristan F1 compared to the control was determined as an increase in the number of fruits by 38–43% and in the weight of fruits by 52–53% per one plant. Moreover, we found an increase in the wet and dry mass of cucumber leaves by 38–40% and 27–32%, respectively, in the leaf surface area by 38–40% and in the leaf water content by 7.3–9.6% [5].

Another approach to soil amendment is based on sol-gel synthesis and applications of carbon, silicon, and metal oxide nanoparticles. We developed new forms of preparations based on water-soluble polyhydroxylated, carboxylated, and amino acid (threonine, hydroxyproline, arginine, methionine, etc.) derivatives of fullerene C_60_ and silica-sols nanostructures with the addition of macro-, microelements, and other physiologically active compounds. A positive effect of treating seeds with these preparations on the plant production process under favorable controlled conditions, under oxidative stress caused by UV-B irradiation and deficiency of soil moisture, as well as in natural conditions of the Leningrad region, was demonstrated [7]. Moreover, treating the surface of barley seeds with silica sols obtained by acidic or alkaline hydrolysis of tetraethoxysilane in excess of water or ethanol in the presence of a number of salts and/or acids was found to affect the plant growth [8] positively.

Owing to its ability to photocatalytic activation of oxygen, one of the most important application areas of TiO_2_ nanoparticles in agriculture is the treatment of crop seeds aimed at accelerating seed germination and crop vegetation, enhancing stress tolerance of seeds and their ability to absorb water and oxygen [9,10]. In [11], the problems of using synthetic titanium dioxide nanoparticles in agrochemistry as biologically active stimulants and micronutrient fertilizers are considered. Functional layers are formed on the surface of the seeds of Chinese cabbage using commercial TiO_2_ nanopowder in the form of aqueous suspensions and in combination with tetraethoxysilane-based silica sols. The positive effect of TiO_2_ nanoparticles on seed germination, the qualitative and quantitative composition of epiphytic microorganisms, and the physiological state of seedlings, which stimulates plant growth and their resistance to stressors, including phytopathogenic microorganisms, was established. The synergistic effect of all factors (silica sol, TiO_2_ nanoparticles, a solution of macro- and microelements, etc.) on the growth and development of plants in the initial period of development was found. In order to increase the efficiency of TiO_2_-based photocatalytic systems, it is necessary to use this substance in the form of stabilized aqueous dispersions or in the form of core-shell nanoparticles with TiO_2_ core and a water-soluble shell. The most promising synthetic method for these core-shell structures consists in modifying titanium dioxide nanoparticle surfaces with covalently grafted chains of water-soluble hydrophilic polymers. In [10], we obtained for the first time core-shell titanium dioxide nanoparticles with a water-soluble shell of polymethacrylic acid and showed their prospects for agricultural applications.

Generally, polymers, especially amphiphilic ones, which are mostly requested in biomedical and agricultural applications, represent the class of materials richest in various nanostructures [12,13].

For the last five years, a number of reviews devoted to applications of natural and synthetic polymers in agriculture have been published. Except for a rather detailed review [2], they were devoted to some particular classes of polymers, which are of great significance in this field, including hydroxyethylcellulose derivatives [14], polyhydroxyalkonoates [15], microbial exopolysaccharides [15,16]. Undisputed leaders in the number of agricultural applications are polysaccharides and hydrogels based on them [17,18,19,20]. It is an explosive development of research in what can be called polymer agriculture that has prompted us to present an overview of the achievements of the last five years in this field. The review is concentrated on polymer applications aimed to optimize, on the one hand, the conditions of plants growing in the root-inhabited zone by using superabsorbent polymer hydrogels for drought control and soil amendments.

## 2. Superabsorbent Polymer Hydrogels

One of the most pressing problems in agriculture is the lack of water for irrigation. Superabsorbent polymers (SAPs) are widely used as containers for water and plant nutrients, especially in arid and semiarid regions, since they significantly improve water usage efficiency [21]. According to a recent review, the application of 100 kg SAP per hectare is the most appropriate rate for increasing seed and dry matter yields and satisfying economic aspects, allowing to increase in cereal seed yield by more than 15% [22].

SAPs are defined as cross-linked polymer networks constituted by water-soluble building blocks [23]. In this review, we will use the classification of superabsorbent polymers or polymer hydrogels, depending on their chemical structure, into synthetic, natural, and semi-synthetic [23].

### 2.1. Formation Mechanisms and Properties of Polymer Hydrogels

Polymer hydrogels are three-dimensional polymer networks consisting of polymer chains cross-linked by means of either chemical or physical methods [24] as shown in Table 1.

Free radical polymerization in the presence of various cross-linking difunctional comonomers (e.g., N,N-methylene bis-acrylamide) and organic compounds (ethylenediamine, epichlorohydrin, citric acid, urea, etc.) is widely used to synthesize polymer hydrogels. Different types of initiation are used, including material initiation thermally (ammonium persulfate, benzoyl peroxide, etc.), photochemically (e.g., 2,2-dimethoxy-2-phenyl-acetophenone), or redox-activated (sodium sulfite and bisulfite, cerium nitrate, etc.) initiators and radiation-induced initiation by γ- or electron beam radiation. It should be noted that non-polymerizable cross-linking agents are superior to difunctional comonomers in safety since no purification of the prepared hydrogel from unreacted toxic acrylic derivatives is required.

Ionotropic gelation due to electrostatic interactions of multivalent metal ions (Ca^2+^, Cu^2+^, Mg^2+^, Fe^2+^, Ba^2+^, Al^3+^, Fe^3+^) with negatively charged polyelectrolytes is an advantageous method of preparation of polymer hydrogels. Polymer hydrogels may also form via the interaction of polycations (e.g., chitosan) with polyanions (e.g., sodium alginate). Moreover, a very important and widely applied method of the synthesis of polymer hydrogels is related to semi-interpenetrating polymer networks formed by chain entanglements of water-soluble linear polymer chains stabilized by hydrogen bonds and Van der Waals interactions [24].

Structural factors determining the properties of chemically cross-linked SAP hydrogels include: (1) the chemical structure of the main chain, (2) the chemical structure of cross-linking chains, (3) the cross-linker nature, (4) the hydrogel neutralization degree, (5) the presence of specially introduced hydrophilic groups [25] as summarized in Table 2.

The choice of the matrix or main chain polymer is based on its properties required for particular applications, including hydrophilicity, water absorption, biodegradability, availability, mechanical stability, etc.

However, the functional SAP properties depend also on the chemical structure of cross-linking polymer chains, i.e., on the choice of the corresponding monomer (Figure 1). The latter should, one hand, polymerize to very high conversion degrees, leaving almost no unreacted toxic residues, and, on the other hand, provide a high amount of hydrophilic groups playing the role of water binding and retention sites. Research of synthetic SAPs mostly deal with polymer hydrogels built from acrylic acid (AA), acrylamide (AM), methacrylic acid (MAA), N,N-dimethylaminoethyl methacrylate (DMAEMA), N,N-dimethylaminopropyl methacrylamide (DMAPMA), 2-acrylamido-2-methylpropane sulfonic acid (AMPS) (Figure 1) cross-linked with such cross-linkers as N,N′-methylene bisacrylamide (MBA), ethylene glycol dimethacrylate (EGDMA), etc. (Figure 2) [23].

The cross-linking agent amount should be optimized to provide a compromise between the mechanical stability of the hydrogel, which increases monotonously with the cross-linker concentration, and water absorption rate, which starts decreasing after some optimal cross-linker concentration is achieved because a tightly cross-linked polymer gel loses the swelling ability.

Hydrogels prepared from polyacrylic or polymethacrylic acids should usually be partially neutralized after the synthesis with sodium or potassium hydroxide, or (meth)acrylic acid should be copolymerized with the corresponding salt during the hydrogel synthesis. Alkalis react with carboxylic groups to form ionic hydrophilic carboxylate groups, and their electrostatic repulsion leads to the hydrogel network expansion, enhancing its water absorption capacity. At some optimal neutralization degree, the electrostatic expansion is balanced by an opposite elastic contraction force, and a further increase in the neutralizing agent concentration causes no effect on the water absorption rate.

The presence of hydrophilic groups in the hydrogel structure is necessary to provide sufficiently high-water absorption and water retention abilities. In this respect, ionic hydrophilic groups are preferable as their affinity to water exceeds that of non-ionic groups. However, interactions of ionic groups with water, in contrast to non-ionic ones, are strongly affected by the surrounding solution salinity, with the water absorption rate decreasing with an increase in the ionic strength of the external solution.

### 2.2. Superabsorbents Based on Synthetic Polymers

#### 2.2.1. SAPs Based on Cross-Linked Copolymers of Partially Neutralized Polyacrylic Acid

One of the most popular synthetic superabsorbent polymers is cross-linked polyacrylamide (PAM), polyacrylic acid (PAA), and its salts. The scheme of the structure of such SAPs, as well as their water-absorbing mechanism via the formation of multiple hydrogen bonds of water molecules with functional groups of a polymer network, is well illustrated in [26] (Figure 3).

The authors [27] synthesized copolymers of AA and sodium acrylate cross-linked with MBA and urea (Figure 4).

The highest water absorbency (909 g/g) was achieved at an equimolar ratio of AA and urea. It was shown that the synthesized SAP has a pronounced effect on maize germination. For different studied types of soils (sandy loam, loam, and paddy soil), adding 0.2% of this SAP promotes seedling and root length (Figure 5). It is important to note that adding 0.5% of SAP causes, on average, no effect on the seedling height (Figure 5A), while the root growth is inhibited (Figure 5B). At an even higher SAP content of 1%, a negative impact for both seedling height and root length is observed (Figure 5A,B).

The most important property of the SAP [27] is its sustained urea release, causing a fertilizing effect on plant growth. It was shown that the time dependence of urea release is almost rectilinear, reaching 3/71% for nutrient N after 40 days. The authors mention that this fact indicates that the N in SAP was released very slowly and could be applicable to crops with a long duration of fertilizer. The mechanism of this urea release involves the swelling of SAP with water followed by gradual hydrolysis of urea from the polymer network [27].

In a number of papers, the effect of the superabsorbent A200 hydrophilic polymer, which is a copolymer of AM, AA, and its potassium salt (Figure 6), on plant growth under drought stress was investigated. Thus, it was found in [28] that the highest growth of *Ficus benjamina* L. ‘Starlight’ was obtained when 1% polymer and a four-day irrigation interval were used.

It was demonstrated in [29] that using Superab A200 under water deficit improved all the traits of corn, causing no effect in the regime of 100% crop evapotranspiration. Mohebi studied the effect of A200 on the growth and physiological responses of date palm seedlings under water deficit conditions [30] and found that using A200 resulted in an increase in superoxide dismutase activity under an 80% irrigation deficit in comparison to full irrigation. Furthermore, it was demonstrated that A200 significantly improved the uptake of mineral nutrients and consequently led to a rise in seedling establishment.

A cross-linked potassium polyacrylate, STOCKOSORB^®^ 660 medium (Evonik Industries, Essen, Germany), was used as a superabsorbent polymer in [31] in two pot experiments using *Zea mays* and *Pinus pinea* as model plants grown in clay and sandy clay loam (SCL) soils with the hydrogel was applied at concentrations of 0, 1, 2, 3 and 4 g/kg soil via either banding or mixing. Its main advantages for agricultural applications are summarized in [32] as its safety and non-toxicity, preventing pollution of agro ecosystems. Moreover, it is rather sensitive to aerobic and anaerobic microbiological degradation and can degrade at rates of 10–15% per year into the water, carbon dioxide, and nitrogen compounds, while its macromolecules are too bulky to be absorbed into plant tissue and, therefore, have zero bioaccumulation potential. The results obtained in [31] showed that hydrogel banding at 0.4% application in SCL soils improves corn’s fresh and dry biomass by 25% and prolongs the survival time of pine seedlings by 90%. Evapotranspiration was greater in soils banded with hydrogel at 0.4% and water use efficiency increased by 10–13% in both soils. An important result is that the hydrogel caused no effect in clay soils, while a 33% increase in water retention at 100 kPa was attained in SCL soils. The same hydrogel was used in [33] to assess its effect on soil physical properties and plant growth parameters using sandy and silty clay loam soils. The results demonstrated that the hydrogel is more efficient in coarse-textured sandy soil in comparison with fine-textured silty clay loam soil. Soil’s physical and hydraulic properties are improved by an increase in the hydrogel concentrations, with the best results obtained at its content of 0.27%. For water use efficiency and corn growth, the best results were obtained at 0.5%.

Another commercially available cross-linked copolymer of acrylamide and potassium acrylate, Aquasorb 3005 KL (SNF FLOERGER, Andrézieux, France), was used to study the water absorbency under loads characteristic of different densities of soil and different depths of application [34,35]. The authors demonstrated a very significant reduction in water absorption capacity by the studied SAP under load. For the highest load of 3.83 kPa, corresponding to the load caused by a 30 cm deep layer of soil of a bulk density of 1.3 g/cm^3^, which is a common value for sandy soils characterized by low water retention capacity, this value decreased to 5.0 g/g from 200 g/g for the control sample without load. For the lowest load in the experiment, which was 0.49 kPa (10 cm deep layer of soil of a bulk density of 0.5 g/cm^3^), this value was 33.0 g/g after 60 min. The soil load caused a prolongation of the swelling time. The time required to reach 63% of maximum absorption capacity increased from 63 min for the control sample to more than 300 min for the largest analyzed load of 3.83 kPa. The authors conclude that the implications of soil load on superabsorbent polymer swelling are crucial for its usage and, thus, for the soil system, and this knowledge should be taken into account for the more effective usage of superabsorbent polymers in agriculture to retain water and to support plant growth [34].

The authors also addressed an important question for agricultural applications of SAPs on how their addition influences the time-dependent flow of water through soil [36]. They found that SAP addition can dramatically hinder the flow rate of water through soil, reducing the permeability by several orders of magnitude and in some cases, causing complete blockage of water infiltration at mass fractions as small as 1%. These results are helpful for evaluations of the optimal proportions and grain sizes of SAPs to use for a given soil in order to achieve both a desirable permeability and an increased water-holding capacity in the plant root zone. The optimization of the SAP concentration and grain size is, thus, crucial in order to benefit from the known ability of synthetic SAPs to increase the soil porosity and permeability, decrease the soil compaction tendency, improve soil drainage, and minimize soil erosion [26,37].

A simple hydraulic model of the mutual interactions between SAP and soil grains, providing more generally applicable and quantitative principles to model SAP-soil permeability in applications, was developed, working remarkably well in describing the variation of the coefficient of permeability with increasing SAP addition, agreeing with the experimentally-measured steady-state values to within a factor of ~3 [36].

In another paper [38], these authors built a purely stochastic model describing the time dependence of swelling pressure of the SAP and soil mixtures, which is important because the swelling pressure of the SAP in soil affects water absorption by SAP, and soil structure. The model predicts rather reasonably the experimental data for three types of SAPs (Figure 7). It is seen from Figure 7 that some deviations of the model predictions from the experimental data are observed only for the highest SAP concentrations. In the authors’ opinion, the obtained results and the proposed model may be applied everywhere where mixtures of SAPs and soils are used to improve plant vegetation conditions [38].

The same collective of authors used a newly developed water-absorbing geocomposite (WAG), consisting of Aquasorb 3005 KL wrapped by a polyester needle-punched nonwoven geotextile made of 100% polyester with wicking abilities and a weight of 150 g/m^2^ to improve soil water retention and soil matric potential [39]. It was found that WAG-treated samples lost more water after heating by a 1100 W power lamp for 72 h and had lower soil matric potential measured by Irrometer field tensiometers at different depths than the control samples. However, after taking into account the water retained in the WAG, it appeared samples wrapped in WAG had more water easily available for plants than the control samples.

The authors of [40] investigated the effect of negatively charged acrylamide copolymers with sodium and potassium salts of AA taken at different ratios on water absorption and release from soil–SAP mixtures during 10 h of drying. It was shown that Water absorption by soil–SAP mixtures significantly improved water retain capacity relative to the soil, the effect becoming more pronounced with an increase in SAP concentration and soil clay content. Moreover, continued absorption of water by SAPs from soils during the first 5 h of drying was proved. One of the SAPs studied in [40] was also found to be extremely useful in decreasing soil bulk density and significantly improving soil porosity and soil water conservation capacity, thereby promoting potato growth in the dry land farming region of Ningxia (China) [41]. The mean potato yield, commodity rate, and net income increased significantly using WOTE SAP (negatively charged acrylamide copolymers with sodium and potassium salts of AA synthesized in the presence of attapulgite clay [40]) at 60 and 90 kg/hectare by 38.2 and 50.5%, 18.5 and 14.1%, and 28.5 and 35.0%, respectively, compared with no SAP. The soil amendment effect of synthetic SAPs found in [41] is due to their ability to increase the number of aggregates with dimensions between 0.25 and 10 mm, which determine soil fertility [26].

#### 2.2.2. SAPs Based on Polyacrylamide and Other Poly(meth)acrylic Derivatives

Hydrogel prepared of poly(N,N-dimethylacrylamide) was also shown to be able to gently releases its absorbed water by a diffusion-driven mechanism when the humidity of the surrounding soil decreases [42].

Russian scientists studied the water retention properties of microgel copolymers of N-isopropylacrylamide (NIPAM) and AA and their interpolyelectrolyte complexes (IPEC) [43]. They showed that copolymers of NIPAM and AA and their anionic IPECs are more efficient in increasing the field water capacity than pure linear PAA and cationic IPECs, respectively, the effect being more pronounced for quartz sand than for loamy sand soil.

A Group of scientists from Voronezh State University (Russia) developed a hydrophilic crosslinked polymer, “Solid water” having supersorbent properties [44], by cross-linking acrylamide dissolved in (meth)acrylic acid or N,N-di(methyl)ethyloxyethyl methacrylate with N,N-methylene-bis-acrylamide, the whole mixture being added to chitosan solution in acetic acid. It was shown that in contrast to Aquasorb (Andrézieux-Bouthéon, France) and other commercially available synthetic hydrogels such as SolidRain (Santiago de Querétaro, Mexico), AquaSource (Yerevan, Armenia), the highest swelling ratio for “Solid water” is observed in alkaline medium, making this SAP the most appropriate for Voronezh leached chernozem [45]. Continuing the studies of these SAPs, the authors loaded them with microelements and humic acids by adding metal salts or humic acids concentrate at the final polymerization stage [46]. It was shown that SAPs containing microelements or humic acids are characterized by a lower swelling degree in comparison with polymers without additives. The efficiency of the synthesized SAPs for agriculture is demonstrated. The addition of the SAPs in the soil in quantities 10–20 kg/hectare leads to barley yield rising up to 25% and soil microbiology activity increases by a factor of 2.

Asamatdinov describes a method of creating hydrogels by cross-linking the hydrolyzed product, which is made from the waste polyacrylonitrile fiber material in the presence of crosslinking agents, making it possible to convert up to 95–99% of the product into a cross-linked polymer, which is able to absorb water in an amount of 700–1000 mL/g [47]. The feedstock cheapness makes it possible to extend the results of the study to a large scale for agricultural applications. In agreement with the data [38], it was shown that the hydrogel swelling degree decreases with increasing soil layer pressure. At pressures between 10 and 100 kPa, almost all water absorbed by the hydrogel is released into the soil due to this hydrogel “oppression” effect. Therefore, when using hydrogels at the rate of 0.1 wt.% in the sand culture of barley, the wilting point difference compared to the control is not registered, i.e., the moisture contained in the hydrogel supports the plant growth as well as the capillaries [47].

The issue of the hydrogel cost is also addressed in [48], where it was shown that the reclaimed sodium polyacrylate hydrogel obtained from used diapers has a large capability to take up and store moisture and is suitable for agricultural utilization. The water absorbency of the reclaimed hydrogel reaches 250 g/g after 100 h at 18 °C.

A rather important problem concerns the low, if any, biodegradability of synthetic SAPs. However, it was shown in [49] that the hydrated technical copolymer of acrylamide and potassium acrylate, containing 5.28% of unpolymerized monomers, is colonized by soil bacteria, and two of those, *Rhizobium radiobacter* 28SG and *Bacillus aryabhattai* 31SG isolated from the watered SAP were found to be able to biodegrade this SAP in pure cultures. They destroyed 25.07 and 41.85 mg of 300 mg of the technical SAP during the 60-day growth in the mineral Burk’s salt medium.

The authors [50] attempted to combine the improved water retention ability of synthetic SAPs with biocompatibility and biodegradability. To this aim, they synthesized novel hydrogels with phosphoric acid-bearing units (Figure 8).

Having introduced BMEP and MEP at the concentrations 0.5–5.0 wt.% and 1.0–4.0 wt.%, respectively, into a hydrogel of polyacrylic acid, they obtained water regain ranging from 105.5 g/g up to 2837.5 g/g.

The destruction of strongly swelling polymer hydrogels and its effect on the water retention capacity of soils was also studied by Russian researchers [51]. It was shown that the loss of hydrogels during three months of the vegetation period because of their biodestruction may exceed 30% of their initial content in irrigated agriculture under arid climatic conditions and more than 10% under humid climatic conditions. The authors conclude that the biodestruction of hydrogels is one of the most important factors decreasing their efficiency under actual soil conditions.

It was shown in [52] that crosslinking PAM via either thermal treatment at 120 °C or gamma-irradiation at 30 kGy leads to the formation of a porous hydrogel structure, which is absent in the initial PAM. The obtained PAM hydrogel samples were loaded with urea, and their release behavior was examined in water. Finally, the effect of hydrogels on the growth of beans (*Vicia faba* L.) was studied.

It is observed that the addition of hydrogels in the amount of 10 g/m^2^ time results in increasing time to wilting of bean plants from 204 h for the control sample to 336 and 384 h for samples planted in the presence of hydrogels of irradiated PAM and heated PAM, respectively. The presence of hydrogel improved the water retention of the sandy soil, the effect being more pronounced for the irradiated PAM than for heated PAM hydrogel [52].

PAM hydrogels are also frequently used as soil amendments to improve water use efficiency and reduce soil salinity. In [53], lab soil column simulation experiments and field experiments were carried out to evaluate these functions of two separate amendments, polyacrylamide-based super absorbent polymer (SAP) and corn straw biochar at different application rates. The simulation experiments showed that both SAP and biochar inhibited the accumulation of soil salinity, with a reduced rate of 9.7–26.3% and 13.5–37.2%, respectively, dependent on the amendment application rates. However, in the case of using SAP, this effect is observed only in the pre-germination and early jointing growth stages of maize, while soil salinization was inhibited throughout the whole growth period in the presence of biochar.

PAM hydrogels combined with plantain peel biochar were found to increase significantly retention of Cd and Zn in the topsoil irrigated by wastewater while to reduce Cd, Cu, and Zn uptake into potato tuber flesh tissue and Cd uptake into tuber peels considerably. The SAP treatment also significantly reduced Cd uptake in the tuber as compared to the control. No acrylamide monomer was detected in tuber flesh and peel samples for all treatments, indicating the possible safe use of SAP and BC in soils to reduce heavy metal leaching and uptake by plants [54].

#### 2.2.3. Composite SAPs with Various Additives and Nanoparticles

In general, applications of synthetic SAPs based on PAA salts face either a challenge of soil compaction due to the accumulation of Na^+^ (sodium polyacrylate hydrogels) or a problem of poor water-absorbing capability due to shrinkage of polyelectrolyte chains in salts solutions and high rigidity of polymer networks (potassium polyacrylate hydrogels cross-linked chemically, mostly by MBA) [55]. One of the ways to solve this problem is by using nanoparticles as physical cross-linkers. It was shown in [56] that the addition of carbonaceous filler in the formulation of SAPs increases water uptake.

In the above-cited work [55], one novel potassium ion-based SAP (K-SAP) with a remarkably increased water uptake was achieved via in-situ copolymerization of acrylic acid and (3-acrylamidopropyl)trimethylammonium chloride, using Ca(OH)_2_ nanoparticles to crosslink polymer chains mainly via chelation (Figure 9).

In this way, the authors managed to prepare a hydrogel. The water-absorbing capacity of 3600 g/g in distilled water, 150 g/g in 0.9 wt.% KCl solution, 130 g/g in 0.9 wt.% NaCl solution, and 2250 g/g in 10 wt.% urea solution [55]. Using calcium carbonate in the form of eggshell, which is an abundant waste material, as a filler for synthetic SAPs makes it possible to solve simultaneously the two problems of reducing the cost of synthetic SAPs and increasing their water absorption capacity [57]. A superabsorbent hydrogel composite based on poly(acrylamide-co-potassium acrylate) (Pam-Ac) as a matrix containing 17 wt.% of chicken eggshell (ES) powder as a filler was synthesized and shown to have an improved gel strength (Figure 10) and the absorption of water and saline solution increased by 100 and 41% (Figure 11).

The interaction between the acrylate and Ca^2+^ ions was proved in FTIR-spectra by a decrease in splitting between symmetric and anti-symmetric carboxylate vibration bands (Figure 12).

The authors conclude that the high values for the swelling, the homogeneous structure, and the good mechanical properties obtained with the incorporation of a relatively high content of a low-cost waste material indicate that this composite is suitable for application in agriculture, providing additionally a more ecologically sound and useful destination for eggshell residue [57].

Nanoclay filler was used in [58] to prepare nanoclay polymer composites by polymerization reaction with 10% AA, AM, 0.9% ammonium per sulfate as initiator, 0.12% MBA as cross-linker loaded with (3%, 5%, 7%, 9%, and 11%) halloysite nanotubes. It was found that the addition of halloysite nanoclay during the polymerization reaction has increased the water sorption capacity of polymer composites, with maximum water sorption observed at 7% clay loading. Furthermore, polymer composite-treated soil has 4% to 6% more available water and a higher soil moisture regime as compared to control (untreated) soil.

Another sample of using halloysite fillers in SAP composites is provided by the paper [59] where a more complex ternary random copolymer consisting of AA, AM, and AMPS monomeric units crosslinked with vinyltrimethoxysilane (VTMS) in the presence of halloysite nanotubes (HNT) was prepared. It was found that equilibrium swelling ratios are correlated with the crosslink densities of nanocomposites, while water retention capacities are governed by storage moduli. A maximum swelling of 537 g/g was observed for the composite with the lowest density, containing 5 wt.% HNT, while the longest water retention was determined for the composite with 1 wt.% HNT, which had the highest storage modulus among all studied samples. Moreover, for this sample, the water release duration of SAPs was prolonged up to 27 days [59].

The rheology investigation suggested a three-phase mechanism of the HNT nanofiller-SAP interaction (Figure 13).

In the first phase, HNTs are incorporated into the nanocomposite structure via secondary interactions, acting as physical cross-linking sites and leading to a maximum storage modulus. In the second phase, the additional amount of the nanofiller induced grafting, resulting in a reduced amount of crosslinking. In the third phase, nanofiller agglomerates form, and the modulus responses become fluctuate [59].

Low-cost composites with high water absorption capacity were prepared by free radical copolymerization of AA, AM and in aqueous media using MBA as a crosslinker, potassium persulfate as initiator, and rice husk (RH) as a filler [60]. The optimized composite containing treated RH showed a maximum water uptake value of 825 g/g in distilled water and good resistance in saline solutions and in the pH range of 6–10. Peanut growth attributes, photosynthetic pigments, and nodulation traits were improved by 60 kg/hectare dosage of the SAP [60].

Highly swelling polymer hydrogels with different monomer compositions were synthesized using MAA, potassium methacrylate, 2-hydroxyethylmethacrylate (HEMA), AM, and MBA as cross-linker by free radical polymerization mechanism [61]. Their TGA study showed remarkable thermal stability in the air, while SEM images revealed the presence of porosity and void volume (Figure 14).

The synthesized SAPs were investigated for their water, urea, potash, and superphosphate uptake and release behavior under ambient conditions. The AMEM-20 composition synthesized at the molar ratio of MAA:AM:HEMA:MBA = 6:1:1:0.5 showed water uptake to the extent of 839 g/g of polymer. In addition, AMEM-20 released the entire absorbed water within 23 days, while 33, 39, and 45% of urea, potash, and superphosphate, respectively, were released at room temperature in a single absorption-desorption cycle. The observed results indicated that AMEM-20 might be used to control the release of water, urea, and potash in agricultural applications [61].

Scientists from China Agricultural University reported the effects of the synergistic application of a polyacrylamide SAP and biofertilizers (*Paenibacillus beijingensis* BJ-18 and *Bacillus* sp. L-56) on plant growth, including wheat and cucumber [62]. Namely, the germination time of wheat seeds treated with biofertilizer + SAP was shortened by 4 days compared with that of the control, while 2 days compared with that of the SAP treatment group. The germination rate of wheat seeds was 22.70% in control, while 70.0% in the BJ-18 + SAP treatment group and 62.0% in the L-56 + SAP treatment group. The results indicated that adding SAP to the biofertilizer of *P. beijingensis* BJ-18 and *Bacillus* sp. L-56 increased the seed germination rate by 47.3% and 39.3%, respectively, compared with the control. Similarly, the biofertilizer + SAP treatment group shortened the germination time of cucumber seed by 2–4 days compared with other treatment groups. Among all the treatments of cucumber, the L-56 + SAP treatment group had the highest germination rate, with a 35.7% increase compared to the L-56 treatment group [62].

The treatments of adding SAP to biofertilizer significantly increased the growth parameters of both wheat and cucumber seedlings, including plant length, fresh weight (FW), and dry weight (DW) of shoots and roots (Figure 15).

The BJ-18 + SAP treatment group showed a maximum increase over control in shoot length (28.2%), root length (42.3%), shoot FW (86.9%), root FW (83.4%), shoot DW (104.9%), and root DW (79.7%) of wheat seedlings. However, no significant effect was found in the BJ-18 treatment group and L-56 treatment group in comparison to the control.

In the case of cucumber plants, the treatment group of L-56 + SAP showed a maximum increase over control in shoot length (47.8%), root length (50.8%), shoot FW (71.9%), root FW (175.7%), shoot DW (68.9%) and root DW (76.9%). All the biometric growth parameters showed an order of L-56 + SAP > BJ-18 + SAP > L-56 > BJ-18 > Control. Similarly, cucumber treated with biofertilizer alone had no significant difference in all the biometric growth parameters as compared to the control.

The study [63] presented a new one-pot up-scalable industrial organic synthesis method of poly(sodium acrylate-co-acrylamide) superabsorbent polymer via partial alkaline hydrolysis of acrylamide using microwave irradiation. The method allows the hydrolysis, polymerization, and gelation to take place in one pot during a very short reaction time (120 s) and with no need to operate under an inert atmosphere. The obtained product with a water absorbency of >1000 g/g was tested for agricultural application as a soil moisture retention agent using onion as an example. An enhancement in the growth of the onion plant is proved in comparison to the growth of the onion without the superabsorbent polymer (Figure 16).

An interesting approach to the synthesis of cheap SAPs by solution polymerization of partially neutralized AA in the presence of cross-linked polyvinylpyrrolidone (PVPP) with tap water as a reaction medium was developed in [64]. The results showed that the maximum water absorption rate of this SAP with the particle size 0.425–0.85 mm was 2297, 333, and 120 g/g in distilled water, tap water, and 0.9% wt.% NaCl solution, respectively. When the SAP with the particle size 0.15–0.25 mm was added to sandy soil at the rate of 0.56%, the saturated moisture of sandy soil increased by 187%, and the infiltration rate decreased by 96.7%. The water evaporation rate decreased by 57.52% and 43.61% at 45 and 25°C, respectively. Thus, there is a perspective for desertification and agriculture, and the suggested way of preparation can be more effectively applied in practical production.

A potassium copolyacrylamide (Super AB A200) added at 0.1%, and 0.5% of the dry weight to the Agromix G6 mixture was used with three irrigation intervals (daily, each alternate day, and every third day) in order to determine whether SAPs can improve water use efficiency and the physiological growth of cherry tomatoes (*Solanum lycopersicum* var. *cerasiforme*) without causing soil toxicity [65]. The mean yield of the experimental cherry tomatoes and water use efficiency were statistically significantly higher where 0.5% SAP was applied compared to where SAP was not applied. No free acrylamide monomer in tomatoes was found up to the detection limit of 5 μg/kg. The SAP used in the study was also proved to be nontoxic. The authors concluded that the application of SAP could increase the yield and water use efficiency of greenhouse-grown cherry tomatoes without toxic side-effects in the soil and in the tomatoes.

#### 2.2.4. Challenges for Agricultural Applications of Synthetic SAPs

To sum up, one may conclude that we have really observed a boom in using synthetic hydrogels in agricultural applications. It is important to emphasize that they are useful not only under drought stress but also under excessive moisturizing conditions (Figure 17).

However, despite the widely reported positive effects of synthetic SAPs on plant growth, the authors of [67] pay attention of researchers to the possible adverse effects of using SAPs in agriculture. They cultivated maize (*Zea mays* L.) seedlings using distilled water or three different SAP hydrogels, sodium polyacrylate (SP), potassium polyacrylate (PP), and sodium polyacrylate embedded with phosphate rock powder (SPP), as growth media. It was shown that after growth in the medium for 3 days, the tips of the leaves of seedlings grown with SP and PP treatment curled up, turned yellow, and a brownish liquid flowed out of the cells. (Figure 18a). The roots were noticeably atrophied and deformed. After growth in the medium for 6 days, the leaves of the seedlings dried up after being treated with SP and PP, and the roots rotted (Figure 18b). However, these effects of the test substances were most likely caused by their very high concentration used in the experiment—455 g of the substance/l of the plant growing container. The effect of the test substances at lower concentrations was not studied by the authors.

However, root and stem-leaf growth of corn seedlings treated with SPP were much better. The authors found that these negative effects of SP and PP treatments are due to a severe disruption of the balance of Ca and K (or Na) in maize seedlings by high concentrations of K or Na in the conventional SAP hydrogels. Much better results obtained under SPP treatments are explained by the presence of Ca-rich additives to the SAPs, as the formulated SPP in their study. These data seem to be very important for the development of improved approaches to the synthesis of better water-saving agents to be applied in agriculture and forestry.

In the context of the challenges of SAPs for agriculture, it is mentioned in the review [68] that there are some changes in soil physical properties, such as soil porosity, bulk density, and structure [69].

There are some disadvantages of synthetic SAPs due to their often lack of biodegradability, non-renewability, and possible toxicity [20,70]. Therefore, their excessive and ill-conceived use can be harmful to plant health and soil fertility and cause environmental pollution [71].

In conclusion of this discussion, it is further emphasized in [26] that although synthetic SAPs, polyacrylamide, are non-toxic to humans, animals, fish, and plants, residual monomer (acrylamide), which can be present in polyacrylamide hydrogels is a neurotoxin dangerous to humans. Pyrolysis of the sodium polyacrylate original sample and the soil/sodium polyacrylate mixed remnants showed that some substances, such as methane sulfonyl chloride, long-chain amides, and esters, were only derived from water-saturated soil treated with SAPs, which might subsequently have negative impacts on the environment and associated agriculture [72].

### 2.3. Superabsorbents Based on Natural Polymers

A serious challenge associated with agricultural intensification is the millions of tons of agricultural waste, such as rice straw, wheat straw, corn straw, bagasse, etc., generated each year [73]. Most agricultural wastes are rich in cellulose, its content varying from 30–40% in straws to 65% in bagasse [74]. In the review [75], 32 different kinds of waste are investigated for chemical modification in order to obtain carboxymethyl cellulose for the production of a superabsorbent hydrogel that can be applied in agriculture. Undoubtedly, the preparation of SAPs based on nanocellulose extracted from agricultural wastes is an important field of research all over the world [73]. As emphasized in [73], using agricultural waste-derived superabsorbent hydrogels as soil amendments to promote agricultural production draws a “farmer-centered” circular process that may maximize net farm income and minimize ecological footprint (Figure 19).

It is known that SAPs based on natural polymers are superior to those based on synthetic polymers in biodegradability but inferior in water uptake and soil amendment efficiency [23]. These basic features of SAPs based on nanocellulose were confirmed in [76], where nanoscale cellulose-based superabsorbents were prepared using the (2,2,6,6-tetramethylpiperidin-1-yl)oxyl (TEMPO)-mediated oxidation method [77,78] from Bleached Eucalyptus Kraft (BEK) pulp (Australian Paper, Maryvale, QLD, Australia) with a chemical composition of cellulose (78.8% ± 0.8), hemicellulose (17.7% ± 0.4), lignin (3.2% ± 0.1), extractives (0.3% ± 0.1), and ash (0.2% ± 0.1) [79]. The properties of the prepared nanocellulose-based SAP were compared to those of a commercial anionic PAM-based SAP. It was shown for clay loam (CL) and sandy (SD) soil that both commercial and nanocellulose SAPs improved the water retention (Figure 20a), decreased soil density (Figure 20b) and increased soil porosity (Figure 20c), but all these effects are more pronounced for the synthetic SAP than for the cellulose-based one [76].

Plant biomass was also found to be the highest in SD soil amended with a commercial superabsorbent. It is emphasized that fast biodegradation of the nanocellulose-based SAP (approximately 50% of its initial mass remains after 5 days of exposure) has a negative effect on CL soil because it enhances the waterlogging effect. Therefore, inhibiting the biodegradation of nanocellulose SAP would be beneficial for agricultural use [76].

Superabsorbent bacterial cellulose (BC) spheres were synthesized from winery byproducts (grape pomace) to serve as natural carriers for such fertilizers as urea [80], using *Komagateibacter medellinensis* bacterial strain ID13488, known for its ability to produce high yields of BC from low-cost carbon sources and acidic culture media [81], thus avoiding previous pH neutralization steps of the raw material due to the acidity of some agro-industrial residues and fermentation environments. It was found that sphere-like BCs were capable of retaining urea up to 375% of their dry weight, rapidly releasing the fertilizer in the presence of water. The authors conclude that sphere-like BCs represent suitable systems with great potential for actual agricultural hazards and grape pomace valorization [80].

The study [82] dealt with the synthesis and characterization of carboxymethylcellulose sodium salt (CMCNa) and hydroxyethyl cellulose (HEC) based biodegradable hydrogels using citric acid (CA) as a crosslinker (Figure 21).

The effects of CA crosslinker on hydrogel swelling properties were studied, and about 600% swelling was observed for the hydrogel synthesized using 2% of CA crosslinker. Cellulose nanocrystals (CNCs) were incorporated into the hydrogel matrix in order to reinforce it and improve mechanical properties, but the desired effect was not achieved because of poor CNCs dispersion within the hydrogel matrix.

It is pointed out in a review applications of SAPs based on polysaccharides in agriculture that despite numerous laboratory samples of such SAPs, only a fraction of them is environmental-friendly, and even a fewer amount may reach the market as commercial products [83]. Nevertheless, researchers all over the world continue to keep making efforts in the development of new and improved SAP formulations.

Novel SAPs prospective for applications in agriculture as soil conditioners were obtained in [84] based on a commercial GENUGEL^®^ *κ*-carrageenan. These SAPs show water sorption capacity from 2400 to 3100%, being highly stable in distilled water for 14 days.

Numerous publications on SAPs based on natural polymers are devoted to chitosan used as a matrix of controlled release of various agrochemicals [85]. Chitosan nanoparticles are produced by various cross-linking methods, including ionic gelation, reverse micellar method, precipitation, sieving, emulsion droplet coalescence, and spray drying [86,87,88,89,90,91].

Ionotropic gelation is emphasized as a simple and mild protocol that uses no chemical cross-linking agents, which reduces the possible toxic side effects of the procedure [92]. However, an overwhelming majority of studies of SAPs based on natural polymers, in general, are devoted to their chemical modifications via either covalent attachment of synthetic polymer chains, chemical cross-linking, or complex formation with other polymers. These systems will be considered in the next section.

### 2.4. Semi-Synthetic Superabsorbent Polymers

Semi-synthetic SAPs are prepared via chemical modifications and cross-linking of natural polymers. Comparative studies of chemical crosslinking reactions applied to different bio-based hydrogels are performed in [93]. In 2006, an SAP was synthesized by grafting 70 mol% neutralized AA onto hydrolyzed collagen, using potassium persulfate as the initiator MBA as the crosslinking agent (Figure 22) [94].

Under optimized reaction conditions, corresponding to the cross-linker concentration of 9.3 mmol/L, the hydrogel reached a swelling ratio of 920 g/g, the water sorption capacity being reduced at higher crosslinking density due to a decrease in the flexibility of the polymer chains and the polymer ability to expand [93].

Another example of using the protein chain of collagen to prepare SAPs is given by the work [95] where the low fertilizer retention problem as a challenge for modern agriculture [96] was addressed via the synthesis of a collagen-g-p(AA-co-AMPS)–Fe(III) slow release fertilizer hydrogel, which is able to absorb a high amount of water (2595 g/g in distilled water and 121 g/g in 0.9 wt.% NaCl solution) and to support an excellent sustained release of phosphoric acid as a phosphorous fertilizer over 30 days.

However, a major part of studies of semi-synthetic SAPs is devoted to chemical modifications and crosslinking of different polysaccharides as the most abundant and easily available source of biopolymers.

#### 2.4.1. Semi-Synthetic SAPs Based on Cellulose

Natural polymers may be applied in the synthesis of SAPs in the form of either commercially available products or, preferably from the ecological and economic viewpoints, various cellulose-rich agricultural wastes. For example, white cabbage biowaste (CB) was used for the synthesis of a nobel SAP via ammonium persulfate-initialized polymerization of a mixture containing AMPS, AA, NaOH, C, and MBA [97]. The results showed that the novel superabsorbent polymer gel exhibits excellent water absorbency, being able to absorb ultrapure water, distilled water, tap water, and 0.9% NaCl solution at the amounts of 1914, 1726, 306, and 114 g/g, respectively. The authors claim that the synthesized SAP can be well adapted to the soil environment and is expected to be an ideal water retention agent.

A method for the synthesis of cellulose anionic hydrogels from cellulose fibers via TEMPO-mediated oxidation followed by dissolution in NaOH/urea and cross-linking by epichlorohydrin (Figure 23) is suggested in [98].

The growth indexes, including the germination rate, root length, shoot length, fresh weight, and dry weight of the seedlings, were investigated (Figure 24).

The hydrogels coded as CH, CH07, and CH15, according to the carboxylate contents of 0, 0.7, and 1.5 mmol/g, were studied. Soil, CH, CH07, CH15, agar hydrogels with agar concentration of 0.3 wt.% (AG03) and 0.6 wt.% (AG06) was used as culture mediums. The results showed the best performance of CH07 as a plant growth regulator both for seed germination and growth. Thus, the prepared cellulose anionic hydrogel is promising for applications in agriculture [98].

Water-soluble cellulose derivatives are often used to prepare SAPs [99,100,101,102,103]. In [99,100,101,102], SAPs based on carboxymethylcellulose (CMC) were synthesized. Particularly, the authors of [99,100] prepared CMC/montmorillonite (MMT) hydrogels via electron beam irradiation [99] or gamma irradiation [90] of CMC/MMT mixed solutions with MBA [99] or AA and MBA [100]. In both cases, SAPs with a high swelling degree of more than 3000% were obtained. However, it was shown that the swelling degree decreases with increasing MMT content in SAPs. The authors emphasize in this context that 1% MMT hydrogel could be considered a water-managing material for agriculture and horticulture in desert and drought-prone areas [90].

In [101], CMC-based superabsorbent hydrogels were synthesized from the CMC sodium salt (CMC-Na), AA, and AMPS (Figure 25) to enhance its water absorbency and salt tolerance for soil-conditioning applications in areas suffering from drought and soil salinization.

Their maximum water absorbency reached 604 and 119% in distilled water and saline water, respectively, and the maximum adsorption of ammonia nitrogen was 30 mg/g. It was shown that the presence of hydrogels could slow down the loss of nutrients in the soil.

In [102], a novel SAP based on CMC) and zeolite with a poly(MAA-co-AM) (PMAA-co-PAM) support network was synthesized. It was shown that the presence of 1.5 wt/% of zeolite decreased the water absorption of the nanocomposites from 33 ± 2 g/g to approximately 22 ± 1 g/g in pure water but slightly in 0.025 mol/L NaCl solutions. Moreover, fertilizer desorption studies confirmed the controlled release behavior, and this trend may be improved by zeolite structure. Thus, the presence of zeolite increased the amount of monobasic potassium phosphate (KH_2_PO_4_) released from 250 to 275 mg of fertilizer per gram of hydrogel. The authors conclude that controlling the water absorption and kinetic properties by adjusting the nanocomposite constituents may increase the applicability of the composites in agriculture, specifically as carrier vehicles in the controlled release of agrochemicals.

Another water-soluble cellulose derivative, hydroxypropyl methylcellulose (HPMC), was used to synthesize HPMC-g-P(AA-co-PASP)/ATP hydrogels containing attapulgite (ATP) via HPMC grafting with acrylic acid, and polyaspartic acid (PASP) produced by the alkaline hydrolysis of polysuccinimide [103]. Under optimum synthesis conditions, the maximum equilibrium absorption of SACs was 1785, 254, and 138 g/g in deionized water, tap water, and NaCl solution (0.9 wt.%), respectively. Furthermore, the reduced leaching of added urea and low water permeability of the treated soils was demonstrated, indicating that the SAPs have the potential for applications in future sustainable agriculture.

In some formulations, cellulose is used together with some natural byproducts. Thus, an epichlorohydrin-crosslinked hydrogel was prepared from a mixture of cellulose and linseed gum solutions dissolved in the NaOH/urea aqueous system. This SAP showed improved water adsorption properties due to the linseed gum component and strengthened porous structure due to the cellulose backbone component [104].

Red liquor produced by acidic sulfite cooking in the paper industry and consisting of lignosulphonate and polysaccharides was successfully used as an extremely cheap feedstock to prepare SAPs with a swelling ratio up to about 280 g/g and water retention ratio of 80% after 24 h at 50 °C [105] (Figure 26).

Different kinds of gum are also popular as natural sources of polysaccharides in the production of semi-synthetic SAPs [106,107]. SAPs based on carboxymethyl tamarind kernel and guar gums were found to be prospective carriers for boric acid as a boron micronutrient, providing its slow release behavior [106,107]. For SAPs prepared in [106], a square root dependence of boron release on time was found, while for SAPs studied in [107], the time required to release 50% of boron (t50) from the boron-loaded SAP was estimated as 96 days, which is almost triple as compared to commercial B fertilizer, boronated single super phosphate (33 days). Gellan gum-containing polyacrylic acid hydrogels significantly improved the moisture properties of plant growth media (clay, sandy, and clay-soil combination), implying that it has great potential in moisture stress agriculture [108]. In [109], a hydrogel was synthesized by grafting Guar gum with methyl methacrylate and crosslinking with polyethylene glycol. The hydrogel was applied in the field of sugarcane crop, and soil moisture content was measured after 20 days of application. A higher moisture content in the area of hydrogel application (28%) was found compared to the area without hydrogel application (10%), making it promising in terms of improving perennial crop productivity and combating moisture stress in agriculture. Grafting polyacrylic acid to a mixture of gum Arabic and agar was used to prepare SAPs, which could keep most moisture up to 27 days as compared to 14 days for soil without hydrogel [110].

The synthesis of lipase enzyme catalyzed biodegradable hydrogel interpenetrating polymer network (hydrogel-IPN) of natural gum polysaccharide, i.e., gum tragacanth (GT) with AM and MAA and their potential application in the delivery of agrochemicals were reported in [111]. The authors showed that the obtained hydrogel-IPN preserves the biodegradable properties of GT, degrading much better in the compost of sewerage discharge plants than in the fresh garden soil. Moreover, soil fertility was not affected by the degradation of hydrogel-IPN. It was demonstrated that the mixing of hydrogel-IPN within the soil enhanced the water retention and holding capacity of soil for a prolonged period as compared to the natural tendency of soil. Hydrogel-IPN was successfully used for the slow and controlled release of urea and calcium nitrate up to 44 h. It was shown for both agrochemicals that the fertilizer release obeys power time dependence with the diffusion exponent “*n*” being in the range between 0.5 and 1.0, indicating the so-called case II diffusion mechanism when the relaxation time of the hydrogel-IPN was slower as compared to the rate of release of both fertilizers., i.e., the gel remains to be in the swollen state after the agrochemical release (Figure 27).

A simple and facile strategy to prepare biodegradable yeast/sodium alginate/poly(vinyl alcohol) superabsorbent microspheres with a diffusion barrier merit was reported in [112]. In the thermo-chemical modification process, the microspheres were soaked in a citric acid solution, semi-dried in a hot air oven at 50 °C, and then cured at 130 °C for 20 min. Such treatment resulted in a denser cross-linked network on the surface of the microspheres, creating a diffusion barrier to slow down the swelling rate, weaken the unexpected leakage of moisture and fertilizer, and limit the release behavior to a sustainable release (Figure 28). As seen from Figure 28, the thermo-chemically modified yeast/SA/PVA microspheres demonstrate a weakened cumulative IBA release in the initial release stage as compared to unmodified yeast/SA/PVA microspheres, the maximum effect being observed for 1:1 sodium alginate-to-citric acid mass ratio.

#### 2.4.2. Semi-Synthetic SAPs Based on Starch

Starch is a natural polysaccharide which is widely used as an initial reagent in the synthesis of SAPs [70,83,113,114,115,116,117]. However, native starch cannot be directly used due to its poor thermo-mechanical properties and higher water absorptivity. Therefore, native starch needs to be modified before its use, and chemical modification techniques are widely employed in industries [113].

The authors [70] modified potato starch via its esterification with succinic anhydride in an aqueous medium with dimethylaminopyridine as a catalyst. It was shown that the modified starch had an equilibrium water absorption capacity of 260 g/g in distilled water, but the water absorption capacity decreased to 20 g/g in 0.9% NaCl solution. Corn seeds were coated with a mixture of modified starch, bentonite, and talc. However, after 14 and 21 days of growth, the dry weights of the corn plants’ roots and shoots did not significantly differ between coated and uncoated seeds, indicating that, after seed emergence and root development, the contribution of the hydrogel coating to the plant’s water requirements becomes negligible.

The paper [114] addresses the problem of salt intolerance of many synthetic SAPs. To this aim, the authors treated water-soluble starch with sulfamic acid in order to introduce sulfonic acid groups, efficiently adsorbing and transporting water molecules in even salt solutions. Inclusion of these modified starch into a hydrogel formed via polymerization of partially neutralized AA and cross-linking by MBA resulted in a decrease in the water, nitrate, ammonium nitrogen, and water-soluble potassium losses by 18.5, 22.8, 88.0, and 63.8%, respectively, as compared to the SPA obtained without starch treatment by sulfamic acid. The authors conclude that synthesized hydrogels could guide the development and wide application of salt-tolerant SAPs in agriculture and horticulture.

In [115], corn starch was grafted with poly(AA-*co*-AM), and the resulting starch-*g*-poly(AA-*co*-AM) superabsorbent polymer was reinforced with natural char nanoparticles (NCNPs) as a physical cross-linker and loaded with urea to synthesize the starch-g-poly(AA-co-AM)/NCNPs/Urea slow release fertilizer finally. It was shown that the prepared starch-*g*-poly(AA-*co*-AM)/NCNPs/Urea has a water absorbency of about 215 g/g and a rather slow fertilizer release (70% of nitrogen was released during 21 days). Moreover, it was shown that using NCNPs doubles the water-retention compared with the neat polymer, reduces the nitrate leaching rate in the soil from 591.8 mg/L for pure urea to 49.5 mg/L for the starch-*g*-poly(AA-*co*-AM)/NCNPs/Urea, and facilitates the degradation process of SAPs as was shown by the soil burial degradation test.

Meng and Ye used AMPS instead of AM and prepared cassava starch-*g*-poly(AA-*co*-AMPS) SAPs [116]. They showed that the introduction of AMPS units improved the storage modulus and crosslinking density of the synthesized starch-*g*-poly(AA-*co*-AMPS), and it was beneficial to form a perfect network structure. The maximum swelling ratio reached 1200 and 90 g/g in distilled water and brine, respectively, resulting from the high ionization constant and hydrophilic ability of AMPS, and improved tolerance to brine. The same authors showed that the addition of 0.8 wt.% of sodium bicarbonate as a porogen during the SAP synthesis makes it possible to increase the water absorbency to 1878 and 119 g/g in distilled water and NaCl aqueous solution, respectively [117].

#### 2.4.3. Semi-Synthetic SAPs Based on Chitosan

Chitosan is a polycationic biopolymer that can be obtained by partial deacetylation of chitin, which is the second most abundant natural polysaccharide. Chitosan is often used as an antiviral, antifungal, and antibacterial agent to protect plants [118]. Chitosan attracts the increasing attention of researchers as a polymer matrix of various hydrogels for sustainable agriculture applications, as is widely reviewed up to now [119,120,121,122,123,124].

The strategies for the synthesis of chitosan-based include chemical cross-linking, grafting of such polymers as polyacrylic acids and polyacrylamide to chitosan backbone, mixing, complex formation, or covalent binding with other polymers. Thus, Russian researchers prepared hydrogels based on N-succinyl chitosan and hyaluronic acid dialdehyde. The dependencies of storage elastic modulus and loss modulus on the load exposure frequency at constant stress were obtained. The obtained values of the complex shear modulus increase with the growth of molecular mass and concentration of N-succinyl chitosan [125]. Sorokin and Lavlinskaya synthesized SAPs containing up to 30% w of N-succinylchitosan and N-maleoylchitosan (Figure 29), adding them to a free radical polymerization system of AA, AM, and MBA [126].

It was shown that the synthesized ecofriendly SAPs enriched in polysaccharide content are characterized by excellent water-absorbing properties. The highest swelling rate of 1144 g/g was achieved for N-succinylchitosan-based polymer, having 2 wt.% of N-succinylchitosan, 0.5 wt.% of cross-linker, and the acrylamide-to-acrylic acid weight ratio of 1:3. Studies of the swelling kinetics showed that the equilibrium swelling is reached during 0–30 min for all synthesized polymers, and equilibrium reaching time rises with an increase in the polysaccharide content. Moreover, swelling is correctly described by the pseudo-second-order model and controlled by the chemical absorption of water molecules.

In [127], a multifunctional microspheric soil conditioner based on chitosan-grafted poly(acrylamide-*co*-acrylic acid)/biochar [CS-*g*-P(AM-*co*-AA)/BC] was prepared. First, the P(AM-*co*-AA) was synthesized and successfully grafted onto CS, and the three-dimensional network structure of microspheres was formed with *N*,*N*-methylenebis(acrylamide) as the cross-linking agent according to the inverse suspension polymerization method. Biochar and urea were encapsulated into the body of microspheres during the polymerization. It was shown that the synthesized CS-*g*-P(AM-*co*-AA)/BC microspheres possess excellent water absorption and retention capabilities, with the release rate of urea being sharply reduced. Moreover, the introduction of biochar gave the microspheres the ability to adsorb heavy metal ions, e.g., Cu^2+^. The authors concluded that the multifunctional soil conditioner held promise for use in soil improvement and agricultural production.

Tomadoni et al. studied the effect of SAPs based on alginate and chitosan on lettuce plants under drought stress [128]. The results showed an 80% increase in fresh weight of the plants grown in substrate supplemented with 5% hydrogels compared to control substrate after 7 days under drought conditions. Moreover, 25-day-old lettuce plants cultivated without SAP (C) and subjected to drought remained very small, and their growth was clearly affected by water deficit (Figure 30). Plants grown in 1% *w*/*w* SAP supplemented substrate were also very small and wilted. However, when the substrate was supplemented with 5% *w*/*w* SAP, plants were noticeably more vigorous and bigger (Figure 30).

Gelatin-chitosan-poly(vinyl alcohol) hydrogels were found to be promising for agricultural applications not only as water containers but also as protective means for plants. It was shown that such hydrogels loaded with inulin are capable of inducing resistance in chili plants against *Phytophthora capsici* [129]. Using chitosan-poly(vinyl alcohol) hydrogels with copper nanoparticles improved the growth of grafted watermelon [130]. Application of these hydrogels with absorbed copper nanoparticles (Cs-PVA-nCu) was found to increase stoma width, primary stem length, and root length by 7%, 8%, and 14%, respectively [130].

It was stressed in [131] that superabsorbent polymers fabricated via grafting polymerization of acrylic acid from chitosan yield materials that suffer from poor mechanical strength. Therefore, the authors performed hybridization of chitosan (CTS) with cellulose (Cell) via chemical bonding using thiourea formaldehyde resin, increasing the flexibility of the produced hybrid, followed by post-graft-polymerization of AA (Figure 31).

The synthesized (CTS/Cell)-g-PAA polymers were shown to be mechanically robust and pH-responsive in a wide pH range due to the presence of chitosan and polyacrylic acid in one homogeneous entity. The obtained structures also possessed greater water absorbency 390, 39.5 g/g in distilled water and saline (0.9 wt.% NaCl solution), respectively, and enhanced retention potential even at elevated temperatures. These novel SAPs were proved to be very efficient devices for the controlled release of NPK fertilizers into the soil. Thus, free NPK-fertilizer applied without SAPs is completely released in about 5 days, while using CTS/Cell)-g-PAA-NPK superabsorbent polymer makes it possible to prolong the release time over 60 days, expanding the use of these SAPs in agriculture and horticultural applications.

## 3. Conclusions

This review presents data from the past five years on the use of polymeric superabsorbent hydrogels in agriculture as water and nutrient storage and retention materials, as well as additives that improve soil properties. The use of synthetic and natural polymeric hydrogels for these purposes is considered. Although natural polymers, such as various polysaccharides, have undoubted advantages related to their biocompatibility, biodegradability, and low cost, they are inferior to synthetic polymers in terms of water absorption and water retention properties. In this regard, the most promising are semi-synthetic polymeric superabsorbents based on natural polymers modified with additives or grafted chains of synthetic polymers, which can combine the advantages of natural and synthetic polymeric hydrogels without their disadvantages. Such semi-synthetic polymers are of great interest for agricultural applications, especially in dry regions, also because they can be used to create systems for the slow release of nutrients into the soil, which are necessary to increase crop yields using environmentally friendly technologies.

## Figures and Tables

**Figure 1 ijms-23-15134-f001:**
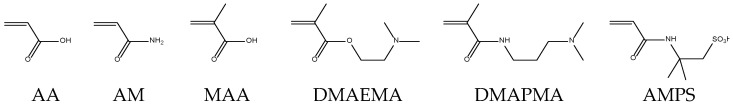
Chemical structures of monomers used as building blocks of synthetic SAPs (AA, AM, MAA, DMAEMA, DMAPMA, AMPS).

**Figure 2 ijms-23-15134-f002:**
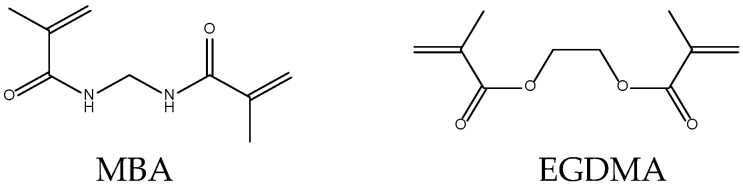
Chemical structures of the most typical cross-linkers are used in synthetic SAPs.

**Figure 3 ijms-23-15134-f003:**
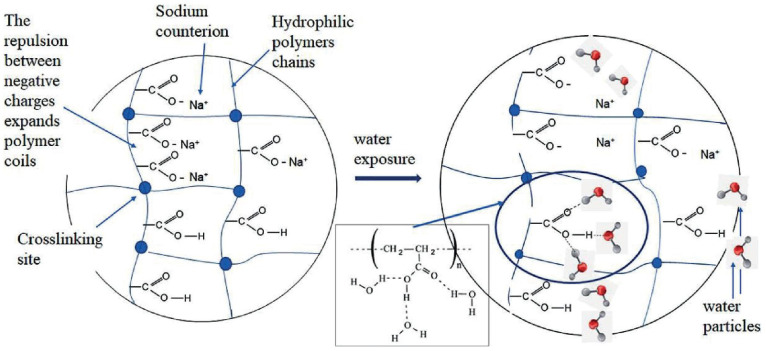
The scheme of the structure of a superabsorbent polymer (sodium-neutralized polyacrylic acid) (Figure 2 from reference [26] with CC-BY attribution).

**Figure 4 ijms-23-15134-f004:**
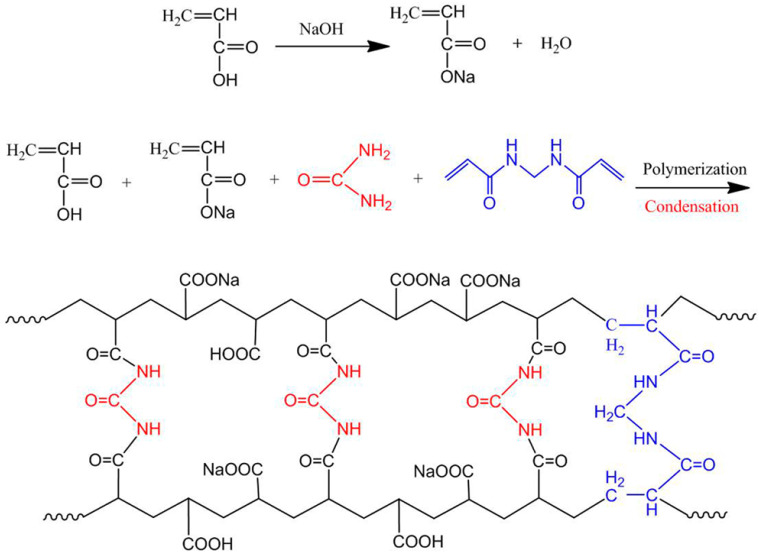
Scheme of the synthesis of a superabsorbent polymer via AA and sodium acrylate copolymerization and cross-linking by MBA and urea [27]. Copyright ACS, 2018. Reproduced with permission from the publisher.

**Figure 5 ijms-23-15134-f005:**
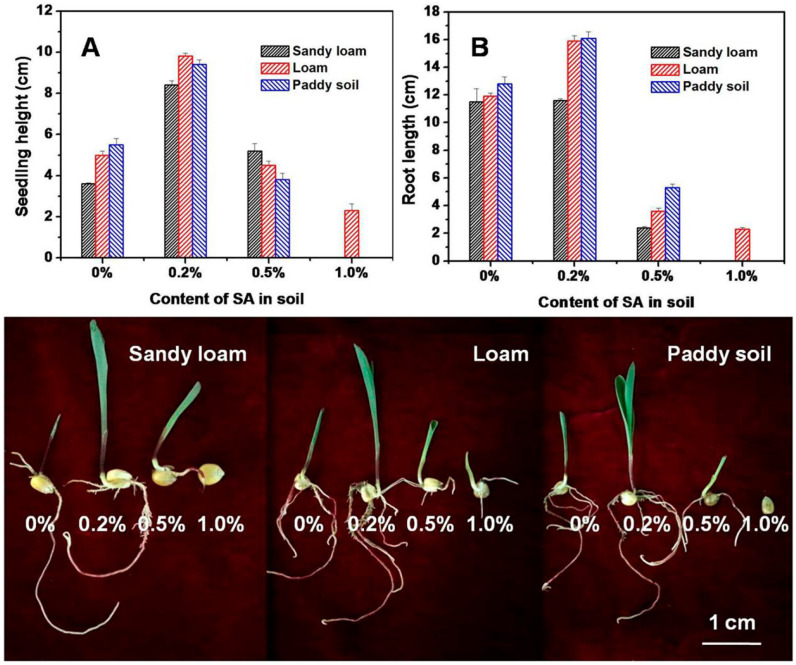
Effect of SAP content in different types of soil on seedling height (**A**) and root length (**B**) in maize [27]. Copyright ACS, 2018. Reproduced with permission from the publisher.

**Figure 6 ijms-23-15134-f006:**
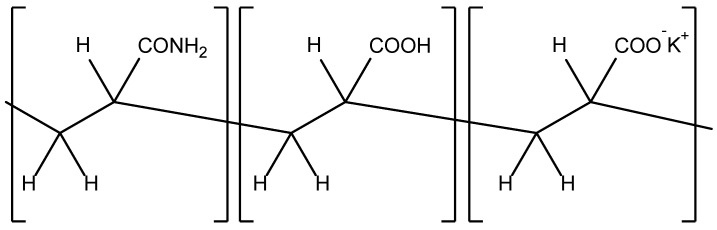
The chemical structure of Superab A200.

**Figure 7 ijms-23-15134-f007:**
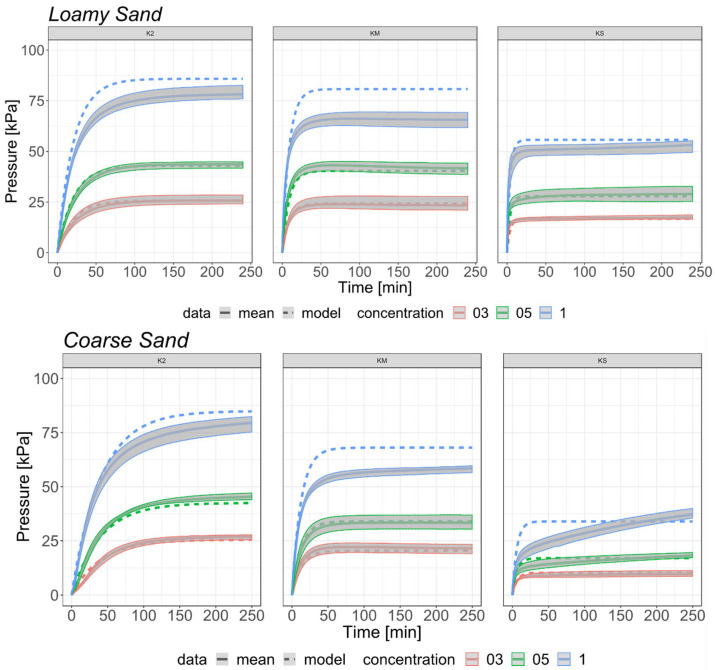
Comparison of the estimated model (dotted line) with measured data (continuous line) of SAP-soil mixtures with a range of variability (a grey area) based on all experiments with Aquasorb 3005 K2 (K2), Aquasorb 3005 KM (KM), and Aquasorb 3005 KS (KS) at the concentrations 0.3% (red), 0.5% (green), and 1% blue with two types of soils (loamy sand and coarse sand) (Figures 5 and 6 from reference [38] with CC-BY attribution).

**Figure 8 ijms-23-15134-f008:**
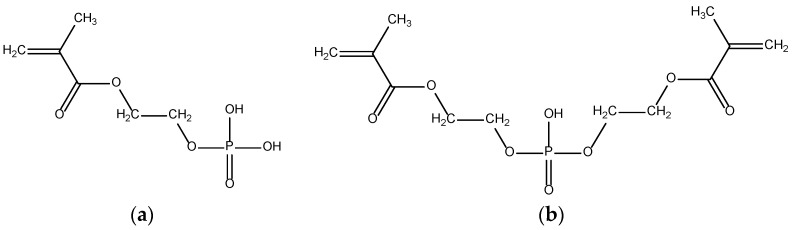
Chemical structures of 2-(methacryloyloxy)ethyl phosphate (MEP) (**a**) and bis [2-(methacryloyloxy)ethyl] phosphate (BMEP) (**b**).

**Figure 9 ijms-23-15134-f009:**
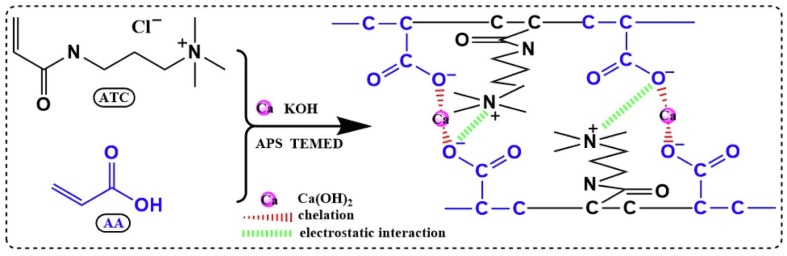
A general cross-linking mechanism between nanoparticles and polymeric chains [55]. Copyright Elsevier, 2022. Reproduced with permission from the publisher.

**Figure 10 ijms-23-15134-f010:**
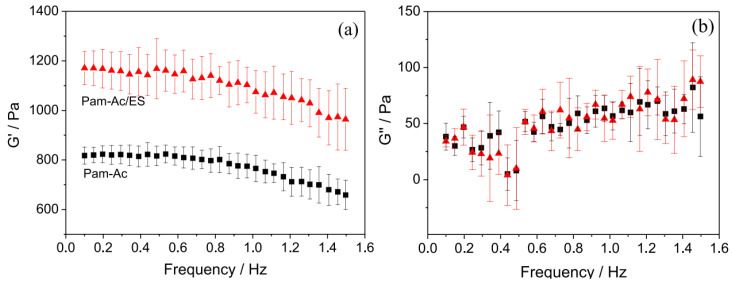
(**a**) Storage modulus (G′) and (**b**) loss modulus (G″) for Pam-Ac and Pam-Ac/ES hydrogels (Figure 6 from reference [57] with CC-BY attribution).

**Figure 11 ijms-23-15134-f011:**
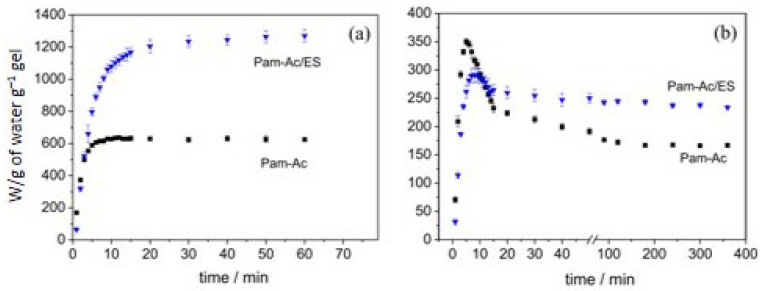
Swelling kinetics plot obtained for hydrogels in (**a**) distilled water and (**b**) salt solution (Figure 7 from reference [57] with CC-BY attribution).

**Figure 12 ijms-23-15134-f012:**
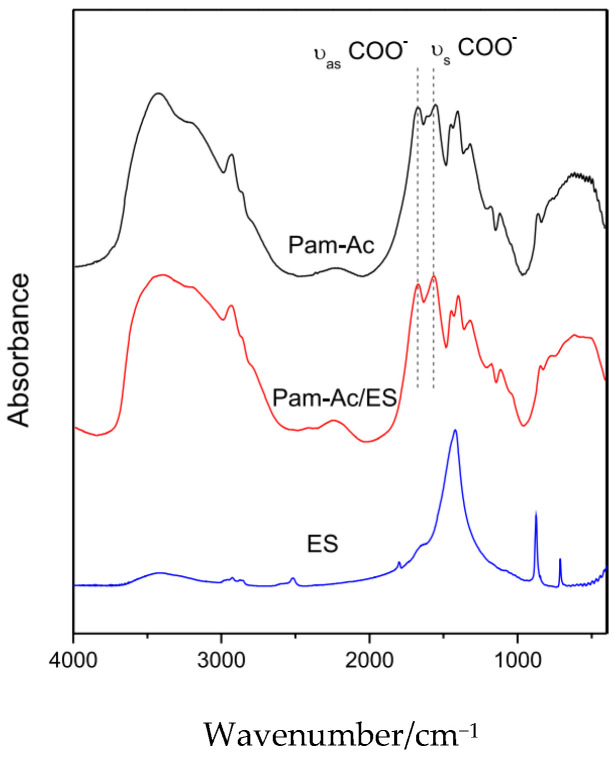
FTIR spectra for eggshell powder (ES), Pam-Ac, and Pam-Ac/ES hydrogels (Figure 3 from reference [57] with CC-BY attribution).

**Figure 13 ijms-23-15134-f013:**
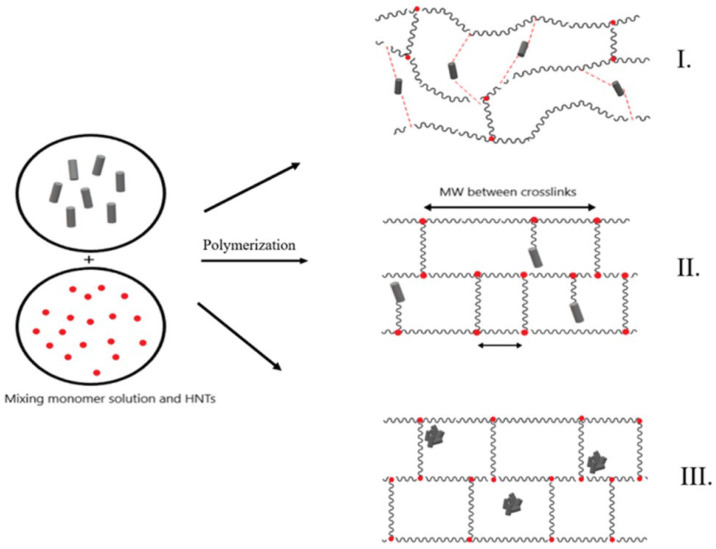
Graphical representation of three different HNT incorporation mechanisms into the SAP nanocomposite structure; intermolecular interactions between the HNT and SAP structure (I), SAP grafting on the HNT surfaces (II), and HNT agglomerated into the SAP structure due to the intramolecular interactions between HNT molecules (III) (Figure 6 from reference [59] with CC-BY attribution).

**Figure 14 ijms-23-15134-f014:**
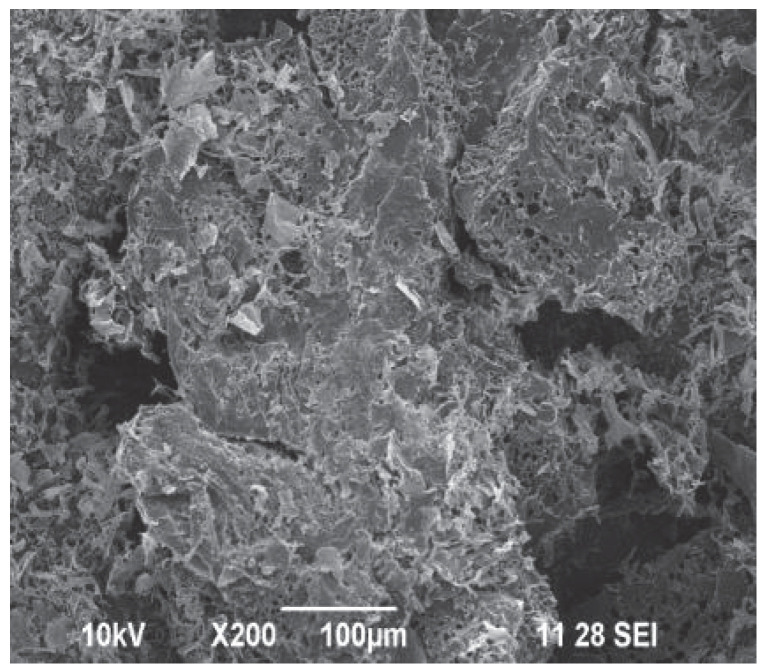
SEM micrograph of lyophilized AMEM-20 synthesized at the molar ratio of MAA:AM:HEMA:MBA = 6:1:1:0.5 [61]. Copyright AIP Publishing, 2021. Reproduced with permission from the publisher.

**Figure 15 ijms-23-15134-f015:**
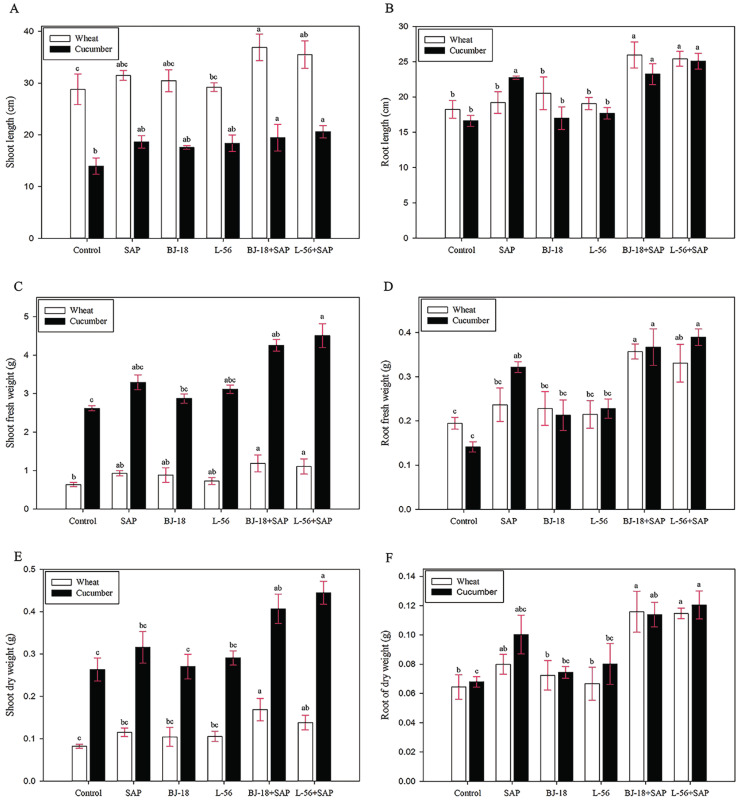
Effects of super absorbent polymer supply on shoot length (**A**), root length (**B**), shoot fresh weight (**C**), fresh root weight (**D**), shoot dry weight (**E**), and root dry weight (**F**) of wheat and cucumber seedlings. Values are given as mean of three independent biological replicates, and bearing different letters (a, b, c) are significantly different from each other according to the least significant difference (LSD) test (*p* < 0:05). The bars represent the standard error. SAP: Super Absorbent Polymer, BJ-18: *P. beijingensis* strain BJ-18, L-56: *Bacillus* sp. strain L-56. (Figure 1 from reference [62] with CC-BY attribution).

**Figure 16 ijms-23-15134-f016:**
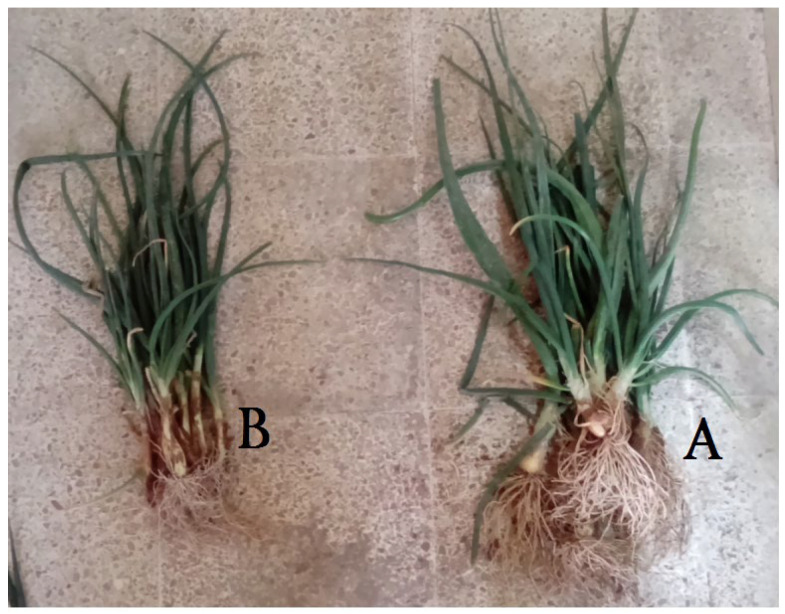
Onion samples were planted (**A**) without adding SAP to the soil and (**B**) without adding SAP (Figure 7 from reference [63] with CC-BY attribution).

**Figure 17 ijms-23-15134-f017:**
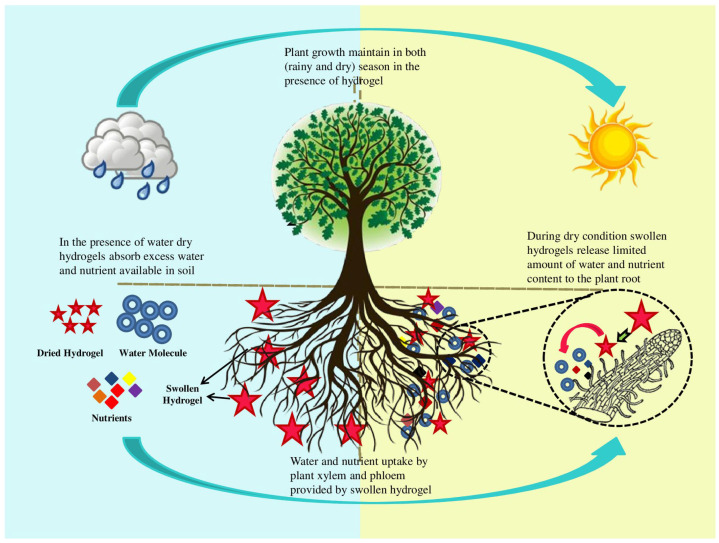
Schematic representation of the slow release of nutrients and water content from hydrogel during different climate conditions [66]. Copyright Elsevier, 2021. Reproduced with permission from the Publisher.

**Figure 18 ijms-23-15134-f018:**
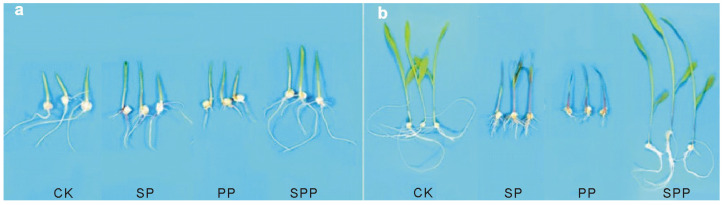
Growth of maize seedlings after 3 (**a**) and 6 (**b**) days treated with distilled water as a control (CK) and three superabsorbent polymer hydrogels, sodium polyacrylate (SP), potassium polyacrylate (PP), and sodium polyacrylate embedded with phosphate rock powder (SPP) [67]. Copyright Elsevier 2017. Reproduced with permission from the publisher.

**Figure 19 ijms-23-15134-f019:**
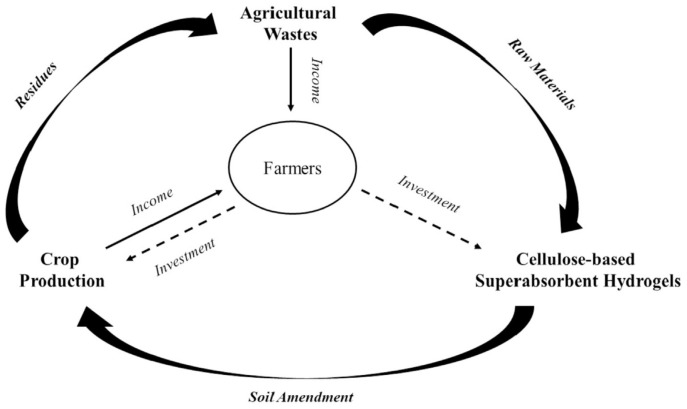
Farmer-centered circular process of producing agricultural waste-derived superabsorbent hydrogels as soil amendments [73]. Copyright Elsevier 2019. Reproduced with permission from the publisher.

**Figure 20 ijms-23-15134-f020:**
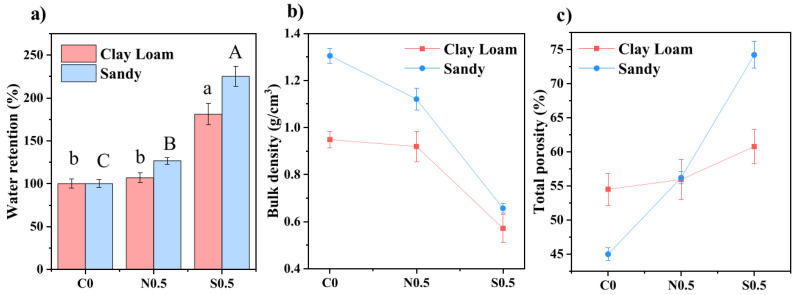
Effect of superabsorbent application on soil properties: (**a**) soil water retention, (**b**) bulk density, and (**c**) soil porosity. Results are reported as mean ± standard deviation (n = 3). Bars with the same letters, lower case (for SD soil) and upper case (for CL soil), were not significantly different at the *p* ≤ 0.05 level (Tukey’s HSD). (1) Here, treatments are referred to as control (C0), nanocellulose at 0.5 wt.% (N0.5), and commercial superabsorbent at 0.5 wt.% (S0.5) [76]. Copyright ACS, 2022. Reproduced with permission from the publisher.

**Figure 21 ijms-23-15134-f021:**
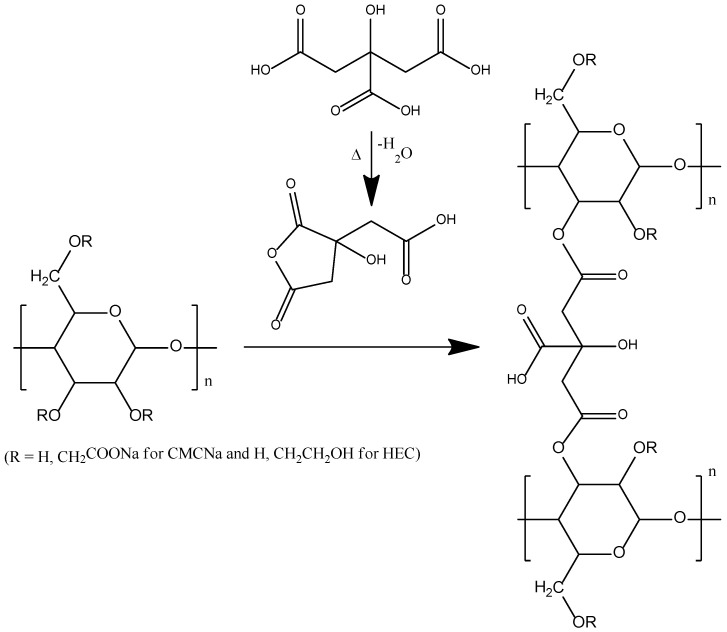
Crosslinking reaction of cellulose molecules with citric acid.

**Figure 22 ijms-23-15134-f022:**
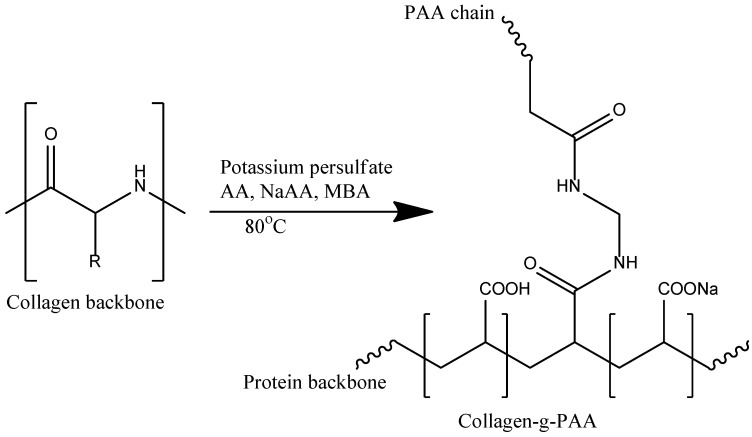
Reaction mechanism in the synthesis of the collagen-g-PAA hydrogel.

**Figure 23 ijms-23-15134-f023:**
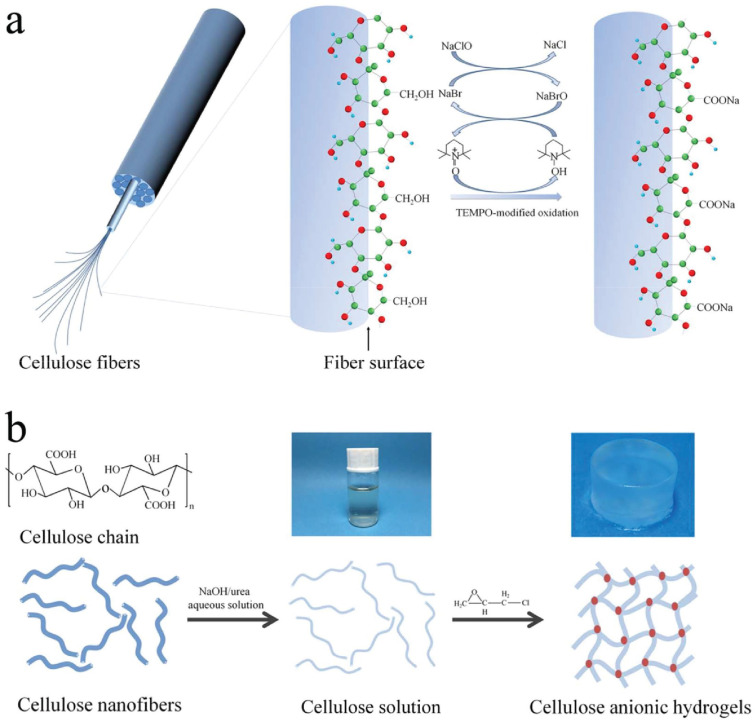
Schematic illustration for the preparation of cellulose nanofibers (**a**) and cellulose anionic hydrogels (**b**) [98]. Copyright ACS, 2017. Reproduced with permission from the publisher.

**Figure 24 ijms-23-15134-f024:**
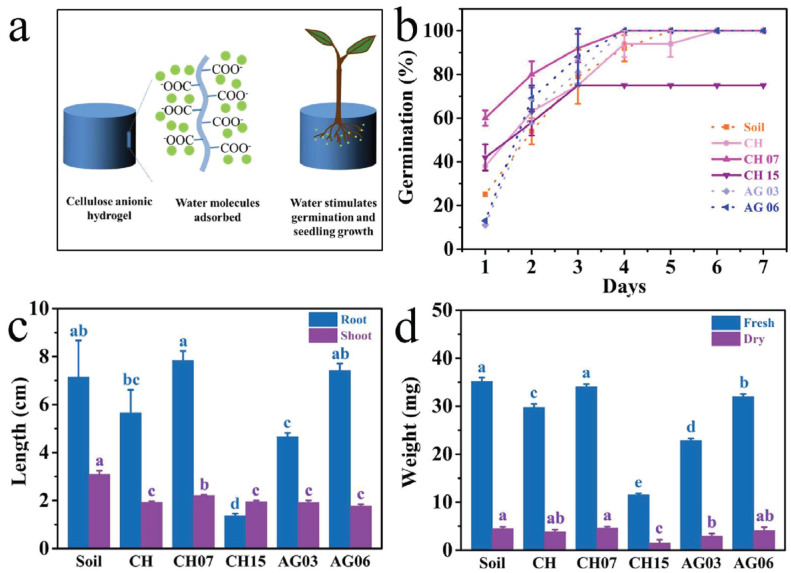
The mechanism of seed germination and seedling growth (**a**), the germination percentage of sesame seeds (**b**), comparison of the length of roots and shoots (**c**), and comparison of the weight of seeding at fresh and dry states (**d**) in soil, CH, CH07, CH15, AG3 and AG6. Statistics of swelling ratios of the hydrogels, seed germination and colony diameters were analyzed using SAS system V8.0 by performing repeated measure 156 analysis of variance (ANOVA). Statistical significance was determined by Duncan’s new multiple range test (*p* < 0.05), and values not sharing a common superscript (a, b, c, d and e) differ significantly [98]. Copyright ACS, 2017. Reproduced with permission from the publisher.

**Figure 25 ijms-23-15134-f025:**
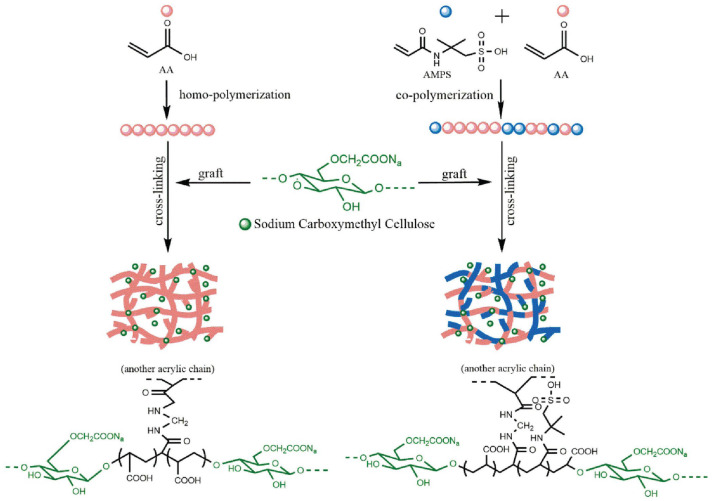
Scheme of the synthesis of salt-tolerant CMC-based SAPs [101]. Copyright Elsevier, 2022. Reproduced with permission from the publisher.

**Figure 26 ijms-23-15134-f026:**
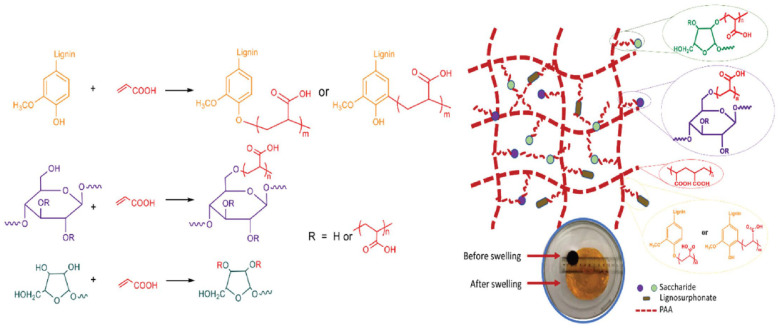
Schematic illustration of the hydrogel network and the chemical structures after the copolymerization of red liquor and acrylic acid [105]. Copyright Elsevier, 2019. Reproduced with permission from the publisher.

**Figure 27 ijms-23-15134-f027:**
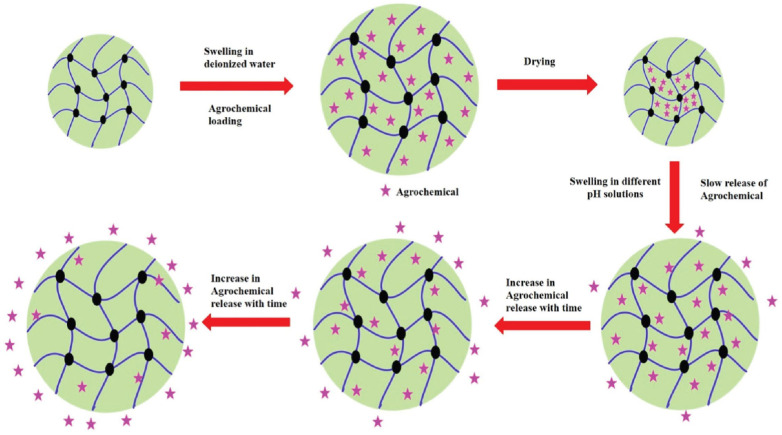
Mechanism of the agrochemical loading and slow release from hydrogel-IPN [111]. Copyright Elsevier, 2019. Reproduced with permission from the publisher.

**Figure 28 ijms-23-15134-f028:**
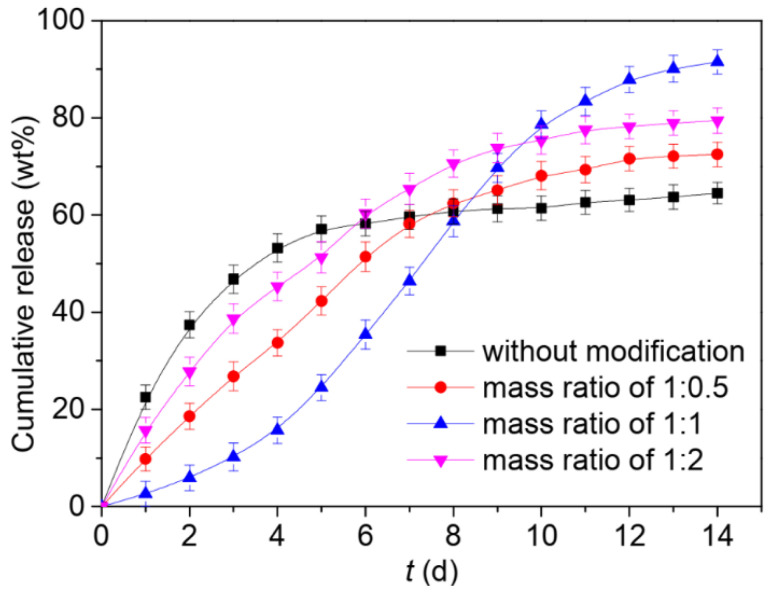
Cumulative release of indole-3-butyric acid (IBA) from the yeast/sodium alginate/poly(vinyl alcohol) microspheres with or without thermo-chemical modification for different sodium alginate to citric acid mass ratios [112]. Copyright ACS, 2017. Reproduced with permission from the publisher.

**Figure 29 ijms-23-15134-f029:**
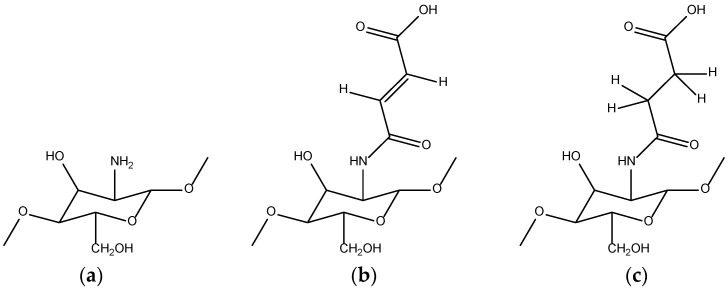
Chemical structures of chitosan (**a**), N-succinylchitosan (**b**), N-maleoylchitosan (**c**).

**Figure 30 ijms-23-15134-f030:**
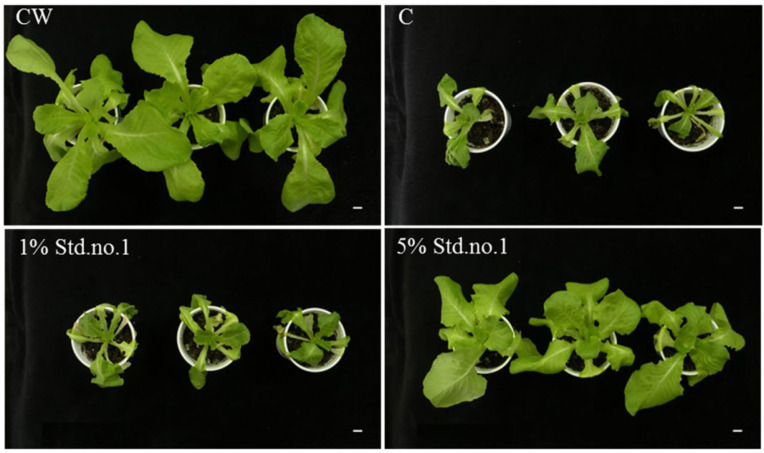
Effect of SAP substrate supplementation on lettuce plants subjected to drought for 6 days; C: substrate without SAP supplementation, 1% *w*/*w* or 5% *w*/*w* SAP-supplemented substrates. Representative plants from each treatment are shown. Bar = 1 cm [128]. Copyright Elsevier, 2020. Reproduced with permission from the publisher.

**Figure 31 ijms-23-15134-f031:**
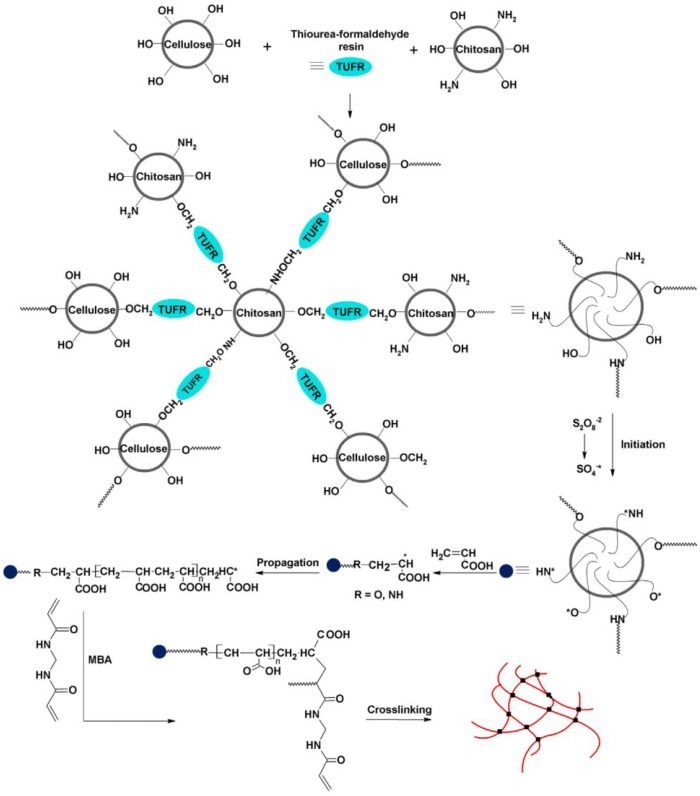
Steps involved in the coupling of cellulose and chitosan into the crosslinked backbone (CTS/Cell) and successive graft polymerization of acrylic acid from this backbone [131]. Copyright Elsevier, 2016. Reproduced with permission from the publisher.

**Table 1 ijms-23-15134-t001:** Formation mechanisms of polymer hydrogels.

Polymer Network Type	Gelation Means	Mechanisms or Agents Involved
Chemical networks	Chemically induced free radical polymerization	Thermal initiation Redox initiation Photo (UV) initiation
	Radiation induced free radical polymerization	γ-radiation Electron beam radiation
	Non-polymerizable cross-linking agents	Ethylenediamine tetraacetic dianhydride Citric acid anhydride Glutaraldehyde Ethylenediamine Epichlorohydrin Urea
Physical networks	Electrostatic interactions	Multivalent metal ions (Ca^2+^, Cu^2+^, Mg^2+^, Fe^2+^, Ba^2+^, Al^3+^, Fe^3+^) Polyelectrolytes
	Semi-interpenetrating network	Chain entanglements via hydrogen bonds and Van der Waals interactions

**Table 2 ijms-23-15134-t002:** Factors affecting the properties of chemically cross-linked superabsorbent polymer hydrogels.

Structural Factor	Available Options		Affected Properties
Chemical structure of the main polymer chain	Synthetic polymers, Natural polymers		Water absorption Biodegradability Biocompatibility Mechanical strength
Monomer, constituting cross-linking chains	Non-ionic	Acrylamide	Water absorption Salinity tolerance Biocompatibility
	Anionic	Acrylic acid Methacrylic acid 2-acrylamido-2-methyl-propane sulfonic acid	
	Cationic	N,N-dimethylaminoethyl methacrylate N,N-dimethylaminopropyl methacrylamide	
Cross-linker structure	N,N′-methylene bisacrylamide Ethylene glycol dimethacrylate Polyethyleneglycol di(meth)acrylate		Elasticity Viscosity Mechanical strength Water absorption
Neutralizing agent	Potassium hydroxide Sodium hydroxide		Water absorption Salinity tolerance Mechanical strength
Hydrophilic groups	Non-ionic	Hydroxyl Amide	Water absorption Salinity tolerance
	Anionic	Carboxyl Sulfonic acid Phosphoric acid	
	Cationic	Amino	

## Data Availability

Not applicable.

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
