# Peer review of "Agricultural Applications of Superabsorbent Polymer Hydrogels"

_ijms, 2022, doi:10.3390/ijms232315134_

Round 1
Reviewer 1 Report
This review paper is very interesting. It is an exhaustive review of all the important work done in the field of polymeric superabsorbent hydrogels. The review covers natural, synthetic, and semi-synthetic hydrogels and comprehensively weighs the positives and negatives. I would like to suggest a few comments to the authors to improve the overall readability and quality of this manuscript.
· I suggest having a table in introduction which lists all the factors that are important for SAP’s. That will make it easier for the reader to understand
· Line 56- Please use full form hectare (ha) to make it easier for the readers
· Line 98- Are you suggesting that higher concentration of SAP negatively impacts seedling growth? Please make this statement clearer to understand
· For section 2.1, I suggest splitting it into smaller sections based on either the chemistry or the factors studied. This will make it easier for the readers as now the text is too large for one section.
Author Response
- I suggest having a table in introduction which lists all the factors that are important for SAP’s. That will make it easier for the reader to understand.
It’s done, the table is included as Table 2 into a new Section 2.1, discussing formation mechanisms and properties of hydrogels (this was also suggested by Reviewer 2).
- Line 56 – Please use full form hectare (ha) to make it easier for the readers.
It’s done.
- Line 98 – Are you suggesting that higher concentration of SAP negatively impacts seedling growth? Please make this statement clearer to understand.
Yes, it follows from the data presented in Figure 5. We reformulated this statement, and, hopefully, it is now clearer: “It is important to note that adding 0.5% of SAP causes, in average, no effect on the seedling height (Figure 5A), while the root growth is inhibited (Figure 5B). At even higher SAP content of 1%, a negative impact for both seedling height and root length is observed (Figure 5A,B).”
- For section 2.1, I suggest splitting it into smaller sections based on either the chemistry or the factors studied. This will make it easier for the readers as now the text is too large for one section.
It’s done. The section, which is now Section 2.2, is subdivided into smaller parts.
Reviewer 2 Report
Comments (ijms-2041016)
This manuscript reviews the applications of superabsorbent polymer hydrogels in agricultural in the recent five years, especially the research progress in drought control and soil amendments in the root-inhabited zone. Although the research on this topic is very meaningful, I suggest that this review manuscript be reorganized and resubmitted in view of the numerous organizational and linguistic errors. Furthermore, there are several points authors need to pay attention for more professionalism. Detailed comments are below:
1. Authors should add their own research publications on this topic.
2. In this manuscript, the authors summarized by material classification. However, the specific properties of polymer hydrogels in agricultural applications are not consistent in the descriptions and comments of various materials, such as water absorption, pesticide release rate, pesticide retention rate, and material expansion rate, etc., which makes it impossible to compare the properties of different materials. It is suggested that the authors present the materials and main properties of polymer hydrogels in a table.
3. This manuscript focuses on the application of polymer hydrogels in agriculture. However, the authors have less elaboration on the formation mechanism of polymer hydrogels, especially the relationship between the structure of polymer hydrogels and their performance in agricultural application. It is suggested that the authors add the mechanism and regularity section.
4. Some spelling mistakes and grammar errors should be polished.
Author Response
- Authors should add their own research publications on this topic.
It’s done (references 1, 4, 5, 6, 7, 8, 10, 11).
- In this manuscript, the authors summarized by material classification. However, the specific properties of polymer hydrogels in agricultural applications are not consistent in the descriptions and comments of various materials, such as water absorption, pesticide release rate, pesticide retention rate, and material expansion rate, etc., which makes it impossible to compare the properties of different materials. It is suggested that the authors present the materials and main properties of polymer hydrogels in a table.
It’s done. The corresponding table is added as Table 2.
- This manuscript focuses on the application of polymer hydrogels in agriculture. However, the authors have less elaboration on the formation mechanism of polymer hydrogels, especially the relationship between the structure of polymer hydrogels and their performance in agricultural application. It is suggested that the authors add the mechanism and regularity section.
It’s done. A new section 2.1 is added.
- Some spelling mistakes and grammar errors should be polished.
English pre-editing by the MDPI service for authors is performed.
Round 2
Reviewer 2 Report
The authors have revised the manuscript in accordance with the review's comments.